

# Kinetics of dimethyl sulfide (DMS) reactions with isoprene-derived Criegee intermediates studied with direct UV absorption

Mei-Tsan Kuo[1], Isabelle Weber[2,6], Christa Fittschen[2], Luc Vereecken[3,4], Jim Jr-Min Lin[1,5]

[1]Institute of Atomic and Molecular Sciences, Academia Sinica, Taipei 10617, Taiwan
[2]Univ. Lille, CNRS, UMR 8522 - PC2A - Physicochimie des Processus de Combustion et de l'Atmosphère, F-59000 Lille, France
[3]Max Planck Institute for Chemistry, Hahn-Meitner-Weg 1, 55128 Mainz, Germany
[4]Institute for Energy and Climate Research, IEK-8: Troposphere, Forschungszentrum Jülich GmbH, 52428 Jülich, Germany
[5]Department of Chemistry, National Taiwan University, Taipei 10617, Taiwan
[6]Present address: Department of Applied Chemistry and Institute of Molecular Science, National Chiao Tung University, Hsinchu 30010, Taiwan

*Correspondence to*: Jim Jr-Min Lin (jimlin@gate.sinica.edu.tw)

**Abstract.** Criegee intermediates (CIs) are formed in the ozonolysis of unsaturated hydrocarbons and play a role in atmospheric chemistry as a non-photolytic OH source or a strong oxidant. Using a relative rate method in an ozonolysis experiment, Newland et al. [Atmos. Chem. Phys., 15, 9521-9536, 2015] reported high reactivity of isoprene-derived Criegee intermediates towards dimethyl sulfide (DMS) relative to that towards $SO_2$ with the ratio of the rate coefficients $k_{DMS+CI} / k_{SO2+CI}$ = 3.5±1.8. Here we reinvestigated the kinetics of DMS reactions with two major Criegee intermediates

formed in isoprene ozonolysis, $CH_2OO$ and methyl vinyl ketone oxide (MVKO). The individual CI was prepared following reported photolytic method with suitable (diiodo) precursors in the presence of $O_2$. The concentration of $CH_2OO$ or MVKO was monitored directly in real time through their intense UV-visible absorption. Our results indicate the reactions of DMS with $CH_2OO$ and MVKO are both very slow; the upper limits of the rate coefficients are 4 orders of magnitude smaller than that reported by Newland et al. These results suggest that the ozonolysis experiment could be complicated such that

interpretation should be careful and these CIs would not oxidize atmospheric DMS at any substantial level.

## 1 Introduction

As a non-photolytic OH source or a strong oxidant, Criegee intermediates (CIs) influence the chemical processes in the troposphere (Nguyen et al., 2016; Novelli et al., 2014; Johnson and Marston, 2008; Atkinson and Aschmann, 1993; Gutbrod et al., 1997; Zhang et al., 2002) and, ultimately, have impact on the formation of secondary aerosols and other pollutants

(Percival et al., 2013; Wang et al., 2016; Meidan et al., 2019). A detailed understanding of CI chemistry under atmospheric conditions is, thus, necessary to be able to accurately predict and describe the evolution of Earth's atmosphere.



However, due to their high reactivity and, hence, short lifetimes, laboratory studies of the reactions of CIs have been challenging. In fact, no direct detection of CIs has been known before Welz et al. reported a novel method to efficiently generate CIs other than through ozonolysis of alkenes (Welz et al., 2012). They utilized (R1) and (R2) to prepare $CH_2OO$

and directly measured the rate coefficients of $CH_2OO$ reactions with $SO_2$ and $NO_2$ by following the time-resolved decay of $CH_2OO$.

$$CH_2I_2 + h\nu \rightarrow CH_2I + I \hspace{4cm} (R1)$$
$$CH_2I + O_2 \rightarrow CH_2OO + I \hspace{3.5cm} (R2, k_{O2})$$

Surprisingly, the obtained rate coefficients are up to $10^4$ times larger than previous results deduced from ozonolysis

experiments, indicating that the ozonolysis experiments could be quite complicated such that reliable kinetic results may be hard to retrieve.

After this pioneering work, the same method has been applied for generation of other CIs, like $CH_3CHOO$ (Taatjes et al., 2013), $(CH_3)_2COO$ (Liu et al., 2014a) and methyl vinyl ketone oxide (MVKO) (Barber et al., 2018), methacrolein oxide (MACRO) (Vansco et al., 2019), etc. These CIs have been identified with various detection methods, like photoionization

mass spectrometry (Taatjes et al., 2013), infrared action (Liu et al., 2014b) and absorption (Su et al., 2013; Lin et al., 2015) spectroscopy, UV-visible spectroscopy (Beames et al., 2013; Sheps, 2013; Liu et al., 2014a; Ting et al., 2014; Smith et al., 2014; Chang et al., 2016), microwave spectroscopy (McCarthy et al., 2013; Nakajima et al., 2015), etc. In addition, utilizing the direct detection of CIs, a number of kinetic investigations of CI reactions, e.g., with $SO_2$ (Huang et al., 2015), water vapor (Chao et al., 2015),  alcohols (Chao et al., 2019), thiols (Li et al., 2019), amines (Chhantyal-Pun et al., 2019), carbonyl

molecules (Taatjes et al., 2012), and organic (Welz et al., 2014) and inorganic (Foreman et al., 2016) acids, etc., have been reported (Lee, 2015; Osborn and Taatjes, 2015; Lin and Chao, 2017; Khan et al., 2018).

Recently, Newland et al. studied the reactivity of CIs with $H_2O$ and, for the first time, with dimethyl sulfide (DMS) in the ozonolysis of isoprene at the EUPHORE simulation chamber facility and found a rapid reaction of CIs with DMS (Newland et al., 2015). A mixture of $CH_2OO$, MVKO and MACRO was generated through ozonolysis of isoprene with a total CI yield

of 0.57 (Newland et al., 2015). The yields of the individual CIs have previously been estimated to be 0.31 for $CH_2OO$, 0.21 for MVKO, and 0.05 for MACRO (Zhang et al., 2002; Nguyen et al., 2016). To determine reaction rates, Newland et al. used a relative rate method and followed the removal of $SO_2$ versus the removal of other reactants. For the reaction CI + DMS relative to the reaction CI + $SO_2$, they obtained a relative rate coefficient of $k_{DMS+CI} / k_{SO2+CI} = 3.5 \pm 1.8$ (Newland et al., 2015). Since the reactions of typical CIs with $SO_2$ are very fast, with rate coefficients on the order of $4\times10^{-11}$ $cm^3s^{-1}$ (Welz et

al., 2012; Lee, 2015; Osborn and Taatjes, 2015; Lin and Chao, 2017; Khan et al., 2018), this result suggests that the reaction of CI + DMS is extremely fast, with a rate coefficient of ca. $10^{-10}$ $cm^3s^{-1}$. This value is extremely large, close to those of the fastest reactions of CIs.

Newland et al. noted, however, that the presented rate coefficients do not correspond to the rates of single elementary reactions but rather describe the general reactivity of CIs towards DMS or $H_2O$ under conditions similar to the atmospheric

boundary layer (Newland et al., 2015).


DMS is the major sulfur containing species in the atmosphere with high abundances in the marine boundary layer (Yvon et al., 1996) but also e.g. in the Amazon basin (Jardine et al., 2015), and has been shown to play an important role in the formation of $SO_2$ and sulfuric acid, which are precursors of sulfide aerosols (Andreae and Crutzen, 1997; Charlson et al., 1987; Faloona, 2009). The results of Newland et al. (Newland et al., 2015) therefore suggest that in regions with high

concentrations of CIs, the CI + DMS reactions will have a comparable impact on the oxidation of DMS, considering the main atmospheric oxidants are OH and $NO_3$ ($k_{DMS+OH}$ = 4.8 × $10^{-12}$ $cm^3$ $s^{-1}$, $k_{DMS+NO3}$ = 1.1 × $10^{-12}$ $cm^3$ $s^{-1}$ (Atkinson et al., 2004)).

Here we report the first direct kinetic study of the reactions of $CH_2OO$ and MVKO, the main CIs formed in the ozonlolysis of isoprene with DMS. CIs have strong UV-visible absorption (Lin and Chao, 2017). For example, $CH_2OO$ and

MVKO absorb strongly (peak cross section $\sigma \geq 1\times10^{-17}$ $cm^2$) in the wavelength ranges of 285−400 nm (Ting et al., 2014; Lewis et al., 2015) and 315−425 nm (Vansco et al., 2018; Caravan et al., 2020) (> 20% of the peak value), respectively. This strong and distinctive absorption has been utilized to probe CIs in a number of kinetic experiments, including their reactions with $SO_2$, water vapor, alcohols, thiols, organic and inorganic acids, carbonyl compounds, alkenes, etc. (Khan et al., 2018; Lin and Chao, 2017; Osborn and Taatjes, 2015; Lee, 2015). In this work, both $CH_2OO$ and MVKO were directly probed in

real time via their strong UV absorption at 340 nm.

Surprisingly, our experimental results do not indicate any significant reactivity of DMS with $CH_2OO$ or MVKO. We therefore propose upper limits of the rate coefficients for these reactions. Implications for atmospheric chemistry are discussed.

## 2. Method

### 2.1 Experimental setup

The experimental setup has been described previously (Chao et al., 2015; Chao et al., 2019). To generate $CH_2OO$ and MVKO, we followed the approaches of Welz et al. (Welz et al., 2012) and Barber et al., respectively. The MVKO formation is through the reaction sequence $ICH_2-CH=C(I)-CH_3 + h\nu \rightarrow CH_3(C_2H_3)CI + I$, $CH_3(C_2H_3)CI + O_2 \rightarrow MVKO + I$, analogue to reactions (R1) and (R2) (Barber et al., 2018). Note that because the MVKO precursor ($ICH_2-CH=C(I)-CH_3$, 1,3-diiodo-2-

butene) does not absorb light at 308 nm (XeCl excimer laser), we used a photolysis laser at 248 nm (KrF excimer laser) for generating MVKO. However, DMS absorbs weakly at 248 nm. We therefore performed additional experiments by photolyzing $CH_2I_2$ at 248 nm to assess the impact of DMS photolysis at 248 nm on the decay of the CIs.

Experiments were conducted in a photolysis reactor (inner diameter: 1.9 cm, effective length: 71 cm). The photolysis laser beam was coupled into and out of the reactor by two long-pass filters (248 nm: Eksma Optics, custom-made 275 nm

long-pass; 308 nm: Semrock LP03-325RE-25) and monitored with an energy meter (Gentec EO, QE25SP-H-MB-D0). The probe light was from a plasma Xe lamp (Energetiq, EQ-99) (Su and Lin, 2013) and directed through the reactor collinearly

with the photolysis beam. It passes through the reactor six times, resulting in an effective absorption path length of ca. 426 cm. After passing through band-pass filters (340 nm, Edmund, #65129, 10 nm bandwidth, OD 4), the probe beam and a reference beam which did not pass through the reactor were both focused on a balanced photodiode detector (Thorlabs,

PDB450A). Output signals were recorded in real time with a high-resolution oscilloscope (LeCroy, HDO4034, 4096 vertical resolution) and averaged for 120 laser shots (repetition rate ~1 Hz). We observed a small time-dependent variation in transmittance even when no precursor was introduced into the reactor. To compensate for this effect, which was caused by the optics and the photolysis laser pulse, we recorded background traces without adding the precursor before and after each set of experiments. The reported data are after background subtraction.

All reactant gas flows were controlled by calibrated mass-flow controllers (Brooks: 5850E, 5800E and Bronkhorst: EL-FLOW prestige) and mixed before entering the reactor. Reactant concentrations were determined prior to mixing of the reactant flows by UV absorption spectroscopy in two separate absorption cells for either DMS (absorption path length 90.4 cm for $[DMS] \leq 1.7 \times 10^{15}$ cm$^{-3}$ or 20.1 cm for $[DMS] \leq 8.1 \times 10^{15}$ cm$^{-3}$) or the respective diiodo precursors (absorption path length 90.4 cm) using the reported absorption cross sections (Sander et al., 2011; Limão-Vieira et al., 2002). However,

because no absorption cross sections for 1,3-diiodo-2-butene have been reported, its absolute concentration cannot be determined. Typical concentration ranges were: $[CH_2I_2]=(0.23–2.54) \times 10^{14}$ cm$^{-3}$, $[O_2]=(3.28–3.30) \times 10^{17}$ cm$^{-3}$, and $[DMS]=(0–8.1) \times 10^{15}$ cm$^{-3}$. We assume ideal gas behavior for the concentration calculation. The majority of the experiments were performed at 300 Torr (N$_2$) and 298 K.

## 2.2 Theoretical methodology

The potential energy surface (PES) of the $CH_2OO$ + DMS reaction was first explored at the M06-2X/cc-pVDZ level of theory (Dunning, 1989; Zhao and Truhlar, 2008), characterizing the geometries and rovibrational characteristics of the reactants, intermediates and transition states for a wide range of potential reaction channels. The pathways found were re-optimized with a larger basis set using M06-2X/aug-cc-pV(T+d)Z, where the triple-zeta basis set is enhanced by tight d-orbitals to improve the description of the sulfur atom bonds (Bell and Wilson, 2004; Dunning et al., 2001). Finally,

CCSD(T)/aug-cc-pVTZ single point energy calculations were performed to obtain more reliable energies (Dunning, 1989; Purvis and Bartlett, 1982). The $T_1$ diagnostics, all $\leq 0.026$ except for $CH_2OO$ (0.042), suggest that the calculations are not affected by strong multi-reference character in intermediates or transition states. The molecular characteristics thus obtained were used in canonical transition state theory (CTST) calculations to derive the temperature-dependent rate coefficient $k(T)$ (Truhlar et al., 1996). All calculations were performed using the Gaussian-09 software suite (Frisch et al., 2010). The

supporting information discusses additional calculations.



## 3 Results and discussion

### 3.1. CH$_2$OO + DMS

Representative time traces of CH$_2$OO absorption recorded at 340±5 nm ($\sigma$ = 1.23×10$^{-17}$ cm$^2$ at 340 nm) (Ammann et al., 2015; Ting et al., 2014) under various [DMS] are depicted in Fig. 1. Similar results but recorded with different initial

concentrations of CH$_2$I$_2$ and/or different photolysis laser fluences are displayed in Figs. S11–S13. At $t$ = 0, CH$_2$OO is generated within 10$^{-5}$ s by photolysis of CH$_2$I$_2$ at 308 nm (nanosecond pulsed laser) (R1) and the fast reaction of CH$_2$I with O$_2$ (R2) ($k_{O2}$ = 1.4×10$^{-12}$ cm$^3$s$^{-1}$ (Eskola et al., 2006); [O$_2$] = 3.3 × 10$^{17}$ cm$^{-3}$). The subsequent decay in absorption is due to the consumption of CH$_2$OO either through reaction with DMS or through other reaction processes, e.g., bimolecular reactions with radical byproducts like I atoms, wall loss, etc. We can see that the decay curves of CH$_2$OO at various [DMS]

are extremely similar to one another, indicating that the reaction of CH$_2$OO + DMS is not significant.

The decay of CH$_2$OO can be well described with an exponential function (R$^2$ > 0.995) (e.g., Fig. 1).

$$[CH_2OO](t) = [CH_2OO]_0 e^{-k_{obs}t} \qquad (1)$$

The fitting error of $k_{obs}$ is less than 1% mostly. Under the conditions of this study, the consumption of CH$_2$OO can be described as

$$-\frac{d[CH_2OO]}{dt} = k_{obs}[CH_2OO] = (k_0 + k_{DMS+CH2OO}[DMS])[CH_2OO] \qquad (2)$$

where $k_0$ represents the sum of the effective rate coefficients for all consumption channels of CH$_2$OO except its reaction with DMS, which is described as the bimolecular rate coefficient $k_{DMS+CH2OO}$.

The CH$_2$OO decay rate coefficients $k_{obs}$ as functions of [DMS] for different photolysis laser fluences are summarized in Fig. 2. At higher laser fluences, more CH$_2$OO and radical byproducts are generated, resulting in shorter CH$_2$OO lifetimes

(see Fig. S7: plot of $k_0$ against [CH$_2$I$_2$]×$I_{308nm}$), similar to previous works (Smith et al., 2016; Li et al., 2020; Zhou et al., 2019). The slopes of the linear fits of Fig. 2 would correspond to $k_{DMS+CH2OO}$ (see Eq. (2)). However, the slope values are quite small, close to our detection limit (Lin et al., 2018). Within experimental uncertainty, $k_{DMS+CH2OO}$ exhibits no clear correlation to the photolysis laser fluence and other experimental conditions like [CH$_2$I$_2$] (see Table S1 and Fig. S9). From a total of 11 experimental data sets (Exp#1−11, Table S1), we inferred an average $k_{DMS+CH2OO}$ = (1.2 ± 1.0)×10$^{-15}$ cm$^3$ s$^{-1}$ (error

bar is one standard deviation of the 11 data points).

### 3.2. Test of the effect of DMS photolysis

Although the absorption cross section of DMS is quite small (1.28×10$^{-20}$ cm$^2$ at 248 nm and <1×10$^{-22}$ cm$^2$ at 308 nm) (Hearn et al., 1990), yet the photolysis of DMS, especially at 248 nm, should be considered. We have performed a quantitative estimation of radical concentrations originating from the photolysis of DMS under the experimental conditions of this work

(page S7) and show the results in Table S4.



In order to reduce the influence of DMS photolysis for the MVKO experiments, which require 248 nm photolysis (see Sect. 3.3), we constraint $[DMS] \leq 1.7\times10^{15}$ cm$^{-3}$ and the laser fluence $I_{248nm} \leq 3.72$ mJ cm$^{-2}$. Then the amount of dissociated [DMS] would be $\leq 1\times10^{11}$ cm$^{-3}$, smaller than the dissociated $[CH_2I_2] \cong 1.2\times10^{12}$ cm$^{-3}$ by an order of magnitude or more.

The expected products of DMS photolysis are $CH_3 + CH_3S$ (Bain et al., 2018), which are less reactive than I atoms or CIs.
Thus, the small amount of dissociated [DMS] would only have a minor effect. And indeed, the results of $CH_2OO+DMS$ reaction obtained with 248 nm photolysis (Figs. S2, S14, Table S2) are very similar to those with 308 nm photolysis (Figs. 2, S1, S11−S13, Table S1), indicating the effect of DMS photolysis is very minor. The values of $k_{DMS+CH2OO}$ obtained with 248 nm photolysis (Table S2) range from $1.6\times10^{-15}$ to $3.2\times10^{-15}$ cm$^3$s$^{-1}$, which are only slightly higher than the results obtained with 308 nm photolysis (see Fig. S9). This indicates that the effect of the DMS photolysis would be on the order of
$(1-3)\times10^{-15}$ cm$^3$s$^{-1}$ for $k_{DMS+CH2OO}$.

### 3.3. MVKO + DMS

Typical absorbance-time profiles of MVKO under various [DMS] ($\leq 1.3\times10^{15}$ cm$^{-3}$) are presented in Fig. 3. When generating MVKO via the reaction of $CH_3(C_2H_3)CI + O_2$ at a high pressure like 300 Torr, the MVKO signal profiles rise slower than those of $CH_2OO$, with the maximum of the MVKO signal being at about 1.5 ms. Based on detailed kinetic and
quantum chemical results, which will be published elsewhere, we have concluded that the slow rise of the MVKO signal is due to the thermal decomposition of an adduct, $CH_3(C_2H_3)CIOO \rightarrow CH_3(C_2H_3)COO + I$ (Lin et al., 2020). See SI (Sect. S3, page S5) for details. This difference is consistent with the fact that MVKO is resonance-stabilized due to the extended conjugation of its vinyl group (Barber et al., 2018) and thus the adduct $CH_3(C_2H_3)CIOO$ is relatively less stable due to disruption of the conjugation. Nevertheless, no significant changes in the absorbance-time profiles of MVKO with varying
[DMS] can be noted (Fig. 3 inset), indicating the reaction of MVKO+DMS is insignificant. In Fig. 3, we can see that the lifetime of MVKO is on the order of 10 ms (i.e., a decay rate coefficient of ca. 100 s$^{-1}$) and the variation of the MVKO signal is insignificant upon adding [DMS]. This indicates that the reaction with DMS only changes, at the most, the MVKO lifetime by a small fraction (< 0.1) (a larger change would cause obvious deviation from the experimental observations of Fig. 3). Thus, $k_{DMS+MVKO}$ can be estimated to be on the order of $(100$ s$^{-1})(0.1)/(1.3\times10^{15}$ cm$^{-3}) \cong 10^{-14}$ cm$^3$ s$^{-1}$. Similar
conclusion can be drawn from additional profiles recorded with different precursor concentrations and photolysis laser fluences or at different pressures (Fig. S15−S17).

To obtain more quantitative values of $k_{DMS+MVKO}$, we performed kinetic analysis and the details are given in SI (Sect. S3); selected results of $k_{obs}$ as functions of [DMS] are presented in Fig. 4. Similar to the $CH_2OO + DMS$ case, the rate coefficients for the reaction MVKO + DMS show no clear dependence on laser fluence and precursor concentration. From a total of 15
experiment sets (Exp#15−29, Table S3), we obtain an average rate coefficient $k_{DMS+MVKO} = (6.2 \pm 3.3)\times10^{-15}$ cm$^3$ s$^{-1}$ (error bar is one standard deviation of the 15 data points). As mentioned above, the MVKO precursor does not absorb light at 308 nm and requires 248 nm photolysis, such that small amounts of DMS would also be photodissociated. However, the above




CH$_2$OO+DMS results indicate that the effect of DMS photolysis in our experiments is minor (on the order of $(1-3)\times10^{-15}$ cm$^3$ s$^{-1}$ for $k_{DMS+CH2OO}$), but may still lead to overestimation of $k_{DMS+MVKO}$. In this regard, the true value of $k_{DMS+MVKO}$ may be

smaller than the above number.

**3.4 Upper limiting rate coefficients and implications for atmospheric modelling**

The experimental values of $k_{DMS+CI}$ (Tables S1 and S3) are quite small, and their standard deviations are comparable to their average values, indicating that the measured $k_{DMS+CI}$ are close to our detection limit. Here we choose the boundary of three standard deviations as the upper limits for $k_{DMS+CI}$, $k_{DMS+CH2OO} \leq 4.2\times10^{-15}$ cm$^3$s$^{-1}$ and $k_{DMS+MVKO} \leq 1.6\times10^{-14}$ cm$^3$s$^{-1}$ (Table

1). From Table 1, we can see that for the reactions of both CIs studied, the upper limits of the rate coefficients for their reactions with DMS, $k_{DMS}$, are much smaller than the literature values of their reactions with SO$_2$, $k_{SO2}$. The resulting ratios $k_{DMS}/k_{SO2}$ are about four orders of magnitude smaller than that reported by Newland et al. (Newland et al., 2015)

The steady-state concentrations of CIs, $[CI]_{ss}$, in the troposphere have not been well established yet (Kim et al., 2015; Khan et al., 2018; Vereecken et al., 2017; Bonn et al., 2014; Boy et al., 2013). Novelli et al. have estimated an average CI

concentration of $5\times10^4$ molecules cm$^{-3}$ (with an order of magnitude uncertainty) for two environments they have investigated (Novelli et al., 2017). Due to fast thermal decomposition (Li et al., 2020; Smith et al., 2016; Vereecken et al., 2017; Stephenson and Lester, 2020) and/or fast reaction with water vapor (Chao et al., 2015; Lee, 2015; Osborn and Taatjes, 2015; Lin and Chao, 2017; Khan et al., 2018), $[CI]_{ss}$ is expected to be low, at least a couple of orders of magnitude lower than the steady-state $[OH]_{ss}$. The small $k_{DMS}$ values obtained in this work imply that these reactions would not compete with the

conventional DMS oxidation pathways like the reactions with OH or NO$_3$, of which both the reactant concentrations and rate coefficients are significantly larger.

Newland et al. performed their experiments on a mixture of 3 CIs (CH$_2$OO, MVKO, MACRO) as resulting from the ozonolysis of isoprene (Newland et al., 2015). The presence of these 3 CIs, however, cannot explain the four orders of magnitude difference to our results. Due to the low yield of MACRO (0.05) compared to the yield of 0.5 for CH$_2$OO +

MVKO (Zhang et al., 2002), it would require an unreasonably large $k_{DMS+MACRO}$ in the order of $10^{-9}$ cm$^3$ s$^{-1}$, to explain the conclusion of Newland et al.

For the determination of the relative rate of the CI + DMS reaction, Newland et al. monitored the consumption of SO$_2$ over a measurement period of up to 60 min until approximately 25% of isoprene was consumed (Newland et al., 2015). Additional uncharacterized sources and/or sinks of SO$_2$ and DMS would lead to a bias in the inferred rate coefficients. A

more likely cause for the discrepancies is differences in chemical compositions of the studied reaction mixtures and, hence, the different impact of side reactions. While our direct measurements and kinetics are very straightforward, the ozonolysis experiments of Newland *et al*. might have been more complex than the authors (Newland et al., 2015) had assumed. For example, one may consider the possibility of converting DMS to SO$_2$ via surface or gas-phase reactions (Chen et al., 2018) under the complicated conditions of isoprene ozonolysis.



### 3.5 Theoretical predictions for the reaction of CH₂OO + DMS

The potential energy surface for $CH_2OO$ + DMS is shown in Figure 5. The reaction proceeds through a pre-reaction complex at −6.0 kcal mol⁻¹ below the free reactants, from which a weakly bonded adduct, $(CH_3)_2SCH_2OO$ at an energy of −2.2 kcal mol⁻¹, can be formed through a submerged TS. At our level of theory, the wavefunction of this adduct converges to a closed-shell species with very strong zwitterionic character. A potential cycloadduct with a 4-membered –SCH₂OO– ring was found to be unstable. Two accessible product-forming transition states were discovered. The first channel starts from the pre-reaction complex, and leads to DMSO + CH₂O by direct transfer of the terminal O-atom of CH₂OO. A high barrier was found, 6.5 kcal mol⁻¹ above the free reactants, leading to a slow reaction despite the predicted strong exothermicity of 79 kcal mol⁻¹ for this channel. The second channel involves the migration of a DMS methyl H-atom to the outer oxygen of the $(CH_3)_2SCH_2OO$ adduct with a barrier of 4.7 kcal mol⁻¹ above the free reactants, endothermically forming $CH_3S(=CH_2)CH_2OOH$ (*i.e.* the methylidene hydroperoxy equivalent of DMSO) with an energy 3.5 kcal mol⁻¹ above the free reactants. No further low-lying reaction channels for this product were found, including formation of $C^•H_2OOH + CH_3SC^•H_2$ which has an energy barrier of ≥ 20 kcal mol⁻¹ at the M06-2X/cc-pVDZ level of theory. We did not examine more exotic CI reaction such as insertion in the DMS C–H bonds, as these are known to have comparatively high barriers. As described in the supporting information, reaction with O₂ appears not competitive, as expected given that all intermediates are closed-shell (zwitterionic) species. For the reactions of DMS with substituted CI (*syn*-CH₃CHOO and *anti*-CH₃CHOO; see supporting information), we found similar complex stability but the adducts are energetically even less favorable, hampering its formation. The most likely fate of the intermediates in the reaction of CI + DMS is thus reformation of the free reactants, with rapid equilibration between free reactants, pre-reaction complex, and adduct. For CH₂OO + DMS, complex and adduct interconvert at rates > 10⁷ s⁻¹ at room temperature (> 4×10⁶ s⁻¹ at 200 K). The lifetime of the complex/adduct with respect to redissociation to the free reactants is estimated to be of the order or microseconds or less at room temperature, assuming a barrierless complexation channel.

The supporting information also describes a set of calculations at a lower level of theory on the catalytic effect of DMS on a set of unimolecular and bimolecular loss processes of CI reactants. We conclude that DMS does not catalyze unimolecular decay of CI, and that DMS does not enhance redissociation of the CI+SO₂ cycloadduct. No information is available on the impact of DMS on the forward reaction rates of CI bimolecular reactions. In the absence of catalytic effects, the observed elementary reaction of CI with DMS must occur through the pathways depicted in Figure 5. The total rate coefficient for product formation, i.e. DMSO or $CH_3S(=CH_2)CH_2OOH$, is predicted at:

$$k(298\ K) = 5.5 \times 10^{-19}\ cm^3\ s^{-1};$$

$$k(200\text{-}450\ K) = 1.34 \times 10^{-44}\ T^{10.28}\ \exp(129\ K/T)\ cm^3\ s^{-1}.$$

Both channels contribute roughly equally at 298 K, with the higher TS being more loose, and the lower TS being more rigid. The $CH_3S(=CH_2)CH_2OOH$ product is intrinsically not very stable, and reverses to the $(CH_3)_2SCH_2OO$ adduct with a rate coefficient ≥ 10¹² s⁻¹, over a very low reverse barrier of 1.3 kcal mol⁻¹. It seems unlikely that this product can undergo any



bimolecular reactions prior to redissociation; reaction with $O_2$ was already found to be very slow. We should then consider that the only stable product effectively formed is $DMSO + CH_2O$, with the following rate coefficient:

$\qquad k_{eff}(298\ K) = 3.1 \times 10^{-19}\ cm^3\ s^{-1};$

$\qquad k_{eff}(200\text{-}450\ K) = 1.34 \times 10^{-26}\ T^{4.40}\ \exp(-2415\ K/T)\ cm^3\ s^{-1}.$

These theoretical rate predictions are in full agreement with the experimental observations on the elementary reactions of CI with DMS.

### 4 Summary

In this work, we present the first direct kinetic study of the reactions of DMS with $CH_2OO$ and MVKO, which are the major CIs formed in the ozonolysis of isoprene. We generate the individual CIs by photolysis of the corresponding diiodo precursors in the presence of $O_2$ and monitored their decay via their strong UV absorption at 340 nm in real time. Our results do not indicate any notable reactivity of DMS with the two CIs studied. We therefore inferred the rate coefficients $k_{DMS+CH2OO} \leq 4.2 \times 10^{-15}\ cm^3 s^{-1}$ and $k_{DMS+MVKO} \leq 1.6 \times 10^{-14}\ cm^3 s^{-1}$. For the reaction of $CH_2OO + DMS$, quantum chemistry

calculation did not find any low-energy reaction pathways, either by direct reaction or by catalysis of unimolecular reactions, and predicted an even smaller rate coefficient of $k_{DMS+CH2OO} = 3.1 \times 10^{-19}\ cm^3 s^{-1}$ at 298 K. Our results indicate that even in regions with high abundance of CIs and high concentrations of DMS, the isoprene-derived CIs will not notably contribute to the oxidation of DMS.

### Acknowledgements

This work is supported by Academia Sinica and Ministry of Science and Technology, Taiwan (MOST 106-2113-M-001-026-MY3; 108-2911-I-001-501(Orchid project)) and French Ministry of Europe and Foreign Affairs through the PHC Orchid project no. 40930 YC. LV is indebted to the Max Planck Graduate Center with the Johannes Gutenberg-Universität Mainz (MPGC), Germany.

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

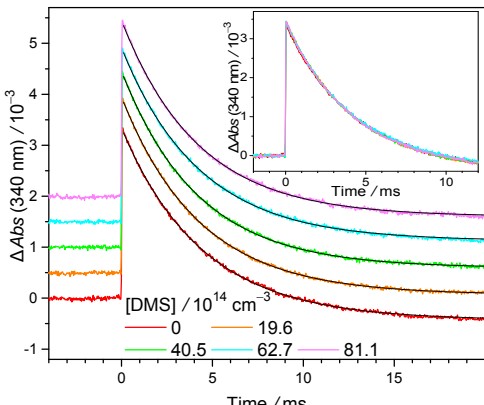

**Figure 1: Representative time traces of CH₂OO absorption recorded at 340±5 nm under various [DMS]. The traces are shifted upward by various amounts for clearer visualization. Smooth black lines are the exponential fit. The photolysis laser (308 nm)**
**pulse defines $t = 0$. The negative baseline (more obvious at long reaction time) is due to depletion of the precursor, CH₂I₂, which absorbs weakly at 340 nm ($\sigma = 8.33\times10^{-19}$ cm²). (Atkinson et al., 2008) This depletion is constant in the probed time window and would not affect the kinetics of CH₂OO. Inset: The profiles without upshifting to show the overlapping. See Exp#1 of Table S1 for detailed experimental conditions.**

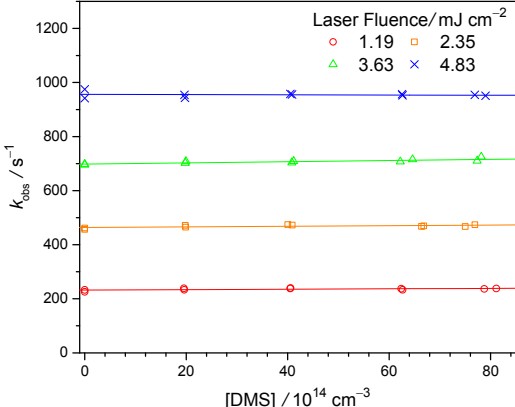

**Figure 2: $k_{obs}$ against [DMS] determined from experiments (Exp#1–4, Table S1) at different photolysis laser fluences $I_{308nm}$; solid lines are linear fits.**

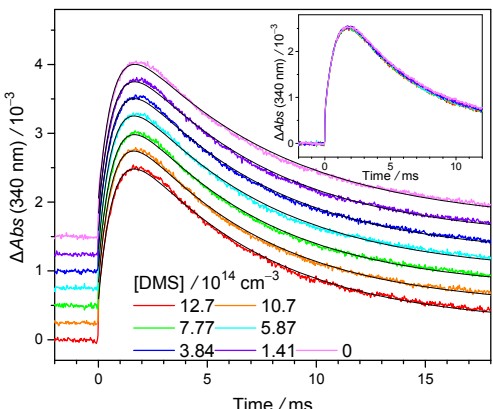

**Figure 3: Representative MVKO absorbance-time profiles recorded at 340 nm under various [DMS] (298 K, 300 Torr, see Exp#16 of Table S3). The profiles are upshifted by various amounts to avoid overlapping. The color lines are experimental data and the smooth black lines are the model fit. Inset: The profiles without upshifting to show the overlapping.**


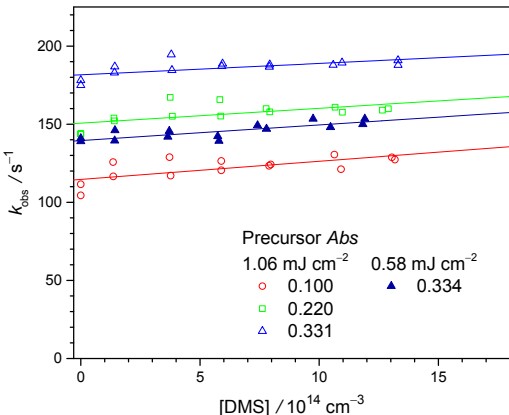

**Figure 4: Plot of the observed decay rate coefficient of MVKO $k_{obs}$ against [DMS] at various laser fluences and precursor absorbances (Exp#15–18). For each data point, the fitting error bar is less than 1% (thus, not shown).**




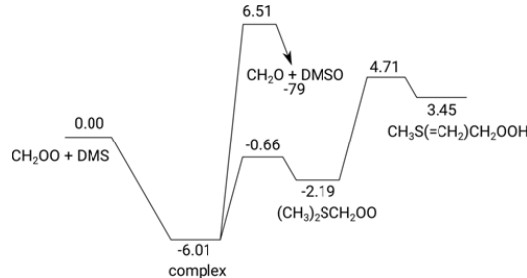


**Figure 5: The potential energy surface of CH$_2$OO + DMS (kcal mol$^{-1}$), based on ZPE-corrected CCSD(T)//M06-2X relative energies.**



**Table 1: Summary of the bimolecular reaction rate coefficients of CI+SO₂ and CI+DMS.**

| CI | $k_{DMS}$ / cm³s⁻¹ | $k_{SO2}$ / cm³s⁻¹ | $k_{DMS}/k_{SO2}$ | Reference |
|---|---|---|---|---|
| CH₂OO | $\leq 4.2\times10^{-15}$ | $3.7\times10^{-11,a}$ | $<1.1\times10^{-4}$ | This work |
| MVKO | $\leq 1.6\times10^{-14}$ | $4.1\times10^{-11,b}$ | $<3.9\times10^{-4}$ | This work |
| CIs | - | - | $3.5\pm1.8$ | Newland et al. 2015 |

$^a$ The average value of (3.4±0.4)x10⁻¹¹ (Stone et al., 2014), (3.5±0.3)x10⁻¹¹ (Liu et al., 2014c), (3.8±0.04)x10⁻¹¹ (Chhantyal-Pun et al., 2015), (3.9±0.7)x10⁻¹¹ (Welz et al., 2012), and (4.1±0.3)x10⁻¹¹ (Sheps, 2013).

$^b$ Caravan et al. 2020.