# Peer review of "Kinetics of dimethyl sulfide (DMS) reactions with isoprene-derived Criegee intermediates studied with direct UV absorption"

_Atmospheric Chemistry and Physics, 2020_

## Short Comment (SC1) · 18 Jun 2020

**Comment on "Kinetics of dimethyl sulfide (DMS) reactions with isoprene-derived Criegee intermediates studied with direct UV absorption" by Kuo et al., 2020.**

**Andrew R, Rickard[1,2], Mike J. Newland[1] and William J. Bloss[3]**

[1] Wolfson Atmospheric Chemistry Laboratories, Department of Chemistry, University of York, York, UK

[2] National Centre for Atmospheric Science (NCAS), University of York, York, UK

[3] School of Geography, Earth and Environmental Sciences, University of Birmingham, B15 2TT, Birmingham, UK

In the work described by Kuo et al. (2020) on the *"Kinetics of dimethyl sulfide (DMS) reactions with isoprene-derived Criegee intermediates studied with direct UV absorption"*, the authors generate SCI from the photolysis of di-iodo precursors, and use a UV absorption technique to demonstrate that the direct reactions of the SCI species $CH_2OO$ and (one conformer of) MVKOO with DMS are slow, with no observable effect of DMS on the rate of SCI loss under their experimental setup, and hence conclude these reactions to be unimportant in the atmosphere. A set of quantum chemical calculations (now added to the original version of the manuscript) on $CH_2OO$ + DMS come to similar conclusions.

The authors focus heavily on the differences between their observations and the only previous experimental study pertinent to these reactions, the chamber work of Newland et al. (2015a) looking at isoprene ozonolysis in the presence of DMS. The directly measured rate constants in the present study, for the reactions of $CH_2OO$ and a subset of the MVKOO stabilised Criegee intermediates, are several orders of magnitude lower than the aggregated relative rate constants determined by Newland et al. (2015a) for all conformers formed in isoprene ozonolysis, using an indirect technique. However, we feel that the main differences between the two complementary studies / techniques are important to interpret the results; and we highlight some considerations here. The earlier Newland et al. (2015a) study looks at the impact of an important atmospheric ozonolysis system as a whole, under representative boundary layer conditions, whereas the present work focuses more on the kinetics of two of the individual component SCI species that are formed in the ozonolysis of isoprene, but synthesised in the laboratory from the photolysis of di-iodo compounds. In this comment we would like to expand on this discussion, looking at how these complementary approaches can be used to give further chemical insight into the relatively complex mechanism of isoprene ozonolysis and its impact on atmospheric chemistry.

The experimental design and the relative rate methodology employed by Newland et al. (2015a) has previously been used to derive SCI yields and kinetic data (i.e. $k(H_2O)$, $k(H_2O)_2$ and $k_d$ (unimolecular dissociation)) for small SCI species formed in a range of atmospherically important ozonolysis systems (namely $CH_2OO$, *syn*-$CH_3CHOO$, *anti*-$CH_3CHOO$ and $(CH_3)_2COO$). These experiments have provided an observational dataset, derived under atmospherically relevant conditions, that is consistent with the well understood general ozonolysis mechanism as well as kinetic data derived from direct literature measurements where individual SCI species are photolytically synthesised from suitable di-iodo precursors (see discussion and references in Newland et al., (2015b)). However, it is important to point out that the isoprene-ozone

system is significantly more complex than these previously studied ozonolysis systems, with a range of different SCI species formed, with different yields and exhibiting different bimolecular and unimolecular kinetics.

Isoprene ozonolysis forms five different initial carbonyl oxides (Scheme 2; Newland et al., (2015a)). The three basic species formed are formaldehyde oxide ($CH_2OO$), methyl vinyl carbonyl oxide (MVKOO) and methacrolein oxide (MACROO). MVKOO and MACROO both have *syn* and *anti* conformers, and each of these can be in either a *cis* or *trans* configuration. Therefore 9 different types and configurations of SCI can be formed in the isoprene ozonolysis system under boundary layer conditions. It is clear from the literature that *syn* and *anti* SCI conformers exhibit significantly different unimolecular and bimolecular kinetics, affecting their atmospheric impacts[1], as discussed in Newland et al. (2015b), and references within.

As noted in Kuo et al. (2020), owing to their high reactivity, and hence short lifetimes, detection of SCI species in the ambient atmosphere has yet to be successful, and direct laboratory studies of SCI kinetics have been challenging until the pioneering work of Taatjes, Percival and co-workers on using photolabile di-iodo precursors that give specific CI in an almost 100% stabilised form (Welz et al., (2012))[2], the synthesis of which have been (until very recently) limited to only the smaller, simpler $C_1$-$C_3$ CI species.

Therefore, in order to investigate the atmospheric impacts of isoprene derived SCI with $SO_2$, water vapour and DMS, Newland et al. (2015a) employed an indirect relative rate technique in which the dependence of $SO_2$ removal in the isoprene-ozone system as a function of water vapour and dimethyl sulfide concentration was used to derive aggregated relative rate data where the combined kinetic effects of the SCI formed are treated as a single "pseudo-SCI" species and as a 2 body system ($CH_2OO$ + CRB-SCI). This experimental approach allows us to assess the atmospheric impact of a range of SCI formed in such an atmospherically important ozonolysis system. It is important to experimentally probe such systems under appropriate boundary layer conditions, which can also be chemically quite complex. The results of such experiments can then be used to drive complementary theoretical investigations as well as direct studies, once the experimental methods are available to synthesise and sensitively detect all of the individual SCI species involved (rather than individual conformers), the results of which can then be compared and contrasted to those from the atmospherically relevant complex system.

5 years on from the original Newland et al. (2015a) study, such methods are now available for MVKOO (Barber et al., 2018, Vansco et al., 2018, Vansco et al., 2019), which have subsequently been employed by Caravan et al. (2020) to look at the kinetics of *syn*-MVKOO with $SO_2$, water vapour and formic acid and in the present study looking at a subset isoprene derived SCI reactions with DMS.
* * *
[1] Note that the authors do not give the relative fractions of *syn* and *anti*-MVKOO formed in the photolysis of 1,3-diiodo-2-butene in the presence of oxygen under the conditions of their experimental set up. It would be useful to include the full distribution of conformers present.

[2] Note that the authors state that "*In fact, no direct detection of CIs has been known before Welz et al. reported a novel method to efficiently generate CIs other than through ozonolysis of alkenes*". This is not in fact the case as Taatjes et al., (2008) directly detected the $CH_2OO$ Criegee, derived from photolytically-initiated Cl oxidation of dimethyl sulfoxide (DMSO), in 2008.

In discussions on the likely causes of the differences seen between the two different studies, the authors state "*Newland et al. monitored the consumption of SO$_2$ over a measurement period of up to 60 min until approximately 25% of isoprene was consumed (Newland et al., 2015). Additional uncharacterized sources and/or sinks of SO$_2$ and DMS would lead to a bias in the inferred rate coefficients. A more likely cause for the discrepancies is differences in chemical compositions of the studied reaction mixtures and, hence, the different impact of side reactions. While our direct measurements and kinetics are very straightforward, the ozonolysis experiments of Newland et al. might have been more complex than the authors (Newland et al., 2015) had assumed. For example, one may consider the possibility of converting DMS to SO$_2$ via surface or gas-phase reactions (Chen et al., 2018) under the complicated conditions of isoprene ozonolysis*"

The potential complexities of the system are addressed in the discussion section of Newland et al. (2015a) and it may be useful for the present study to reflect aspects of this - as discussed in the uncertainties section of Newland et al. (2015a) – annotated with additional points in bold below: *"It is important to note that no constraints regarding the products of the proposed DMS + SCI reaction were obtaine*d; *OH reaction with DMS is complex, proceeding through both abstraction* **(e.g. Veres et al., 2020)** *and addition/complex formation channels, the latter rendered partially irreversible under atmospheric conditions through subsequent reaction with O$_2$ (Sander et al., 2011). The observed behaviour of the experiments is not consistent with* **"non-reactive"** *reversible complex formation dominating the SCI-DMS system under the conditions used; however it is possible that* **"reactive"** *decomposition of such a complex, with DMS reformation* **(i.e. net isomerisation of the SCI)**, *or its further* **"catalytic"** *reaction (e.g. with SO$_2$, analogous to the secondary ozonide mechanism proposed by Hatakeyama et al., 1986), would be consistent with the observed data, and also imply that the reaction may not lead to net DMS removal. Time-resolved laboratory measurements and product studies are needed to provide a test of this mechanistic possibility.*" It is unlikely that heterogeneous chemistry is playing a role (as suggested by the Kuo et al.) given the experimental conditions employed (little aerosol formed, very low surface to volume ratio of $\sim$ 1 m$^{-1}$ of the chamber limited any dark wall reactions – it may be instructive to compare this to the ratio for the laboratory set-up).

In conclusion, the chamber experiments performed by Newland et al. (2015a), under atmospherically relevant conditions, show a clear dependence of SO$_2$ removal in the isoprene + ozone system as a function of dimethyl sulfide concentration. Under the carefully designed (but chemically complex) conditions employed, this behaviour was interpreted to arise from a rapid reaction between isoprene-derived SCIs and DMS. However, in the light of the current study by Kuo et al. (2020) looking directly at individual CH$_2$OO and MVKOO reactions with DMS, coupled to the theoretical work presented on the CH$_2$OO + DMS system, it would appear that this observation may not be the result of a direct reaction with a stabilised Criegee intermediate – at least for those conformers formed in the Kuo et al. experiments. One explanation of this observation is that DMS could be acting to catalyze certain reactions, either chemically (by acting as a transfer intermediate) or energetically (e.g. energy release in complexation or lowering barriers by complexation without being a reaction partner). Some discussion on this is now given in the additional theoretical section of the

supplementary material to Kuo et al., (2020). Clearly there is still more work needed on the detailed atmospheric chemistry of isoprene ozonolysis. Higher level theory quantum chemical calculations and repeat experiments of isoprene ozonolysis in the presence of DMS, including the exploration of conformer-dependent reactivity, would be very timely and may reveal previously unidentified chemical pathways.

Kuo et al., *Atmos. Chem. Phys. Discuss.,* 2020, https://doi.org/10.5194/acp-2020-484

Newland et al., *Atmos. Chem. Phys.,* 15, 9521–9536, 2015a.  doi:10.5194/acp-15-9521-2015

Newland et al., *Phys. Chem. Chem. Phys.,* 17, 4076–4088, 2015b. DOI: 10.1039/c4cp04186k

Barber et al., J. Am. Chem. Soc. 2018, 140, 10866-10880

Vansco et al., J. Chem. Phys. 2018, 149, 244309

Vansco et al., J. Am. Chem. Soc. 2019, 141, 15058-15069

Caravan et al.,  2020,  www.pnas.org/cgi/doi/10.1073/pnas.1916711117

Veres et al., PNAS, 117, 4505–4510, 2020, www.pnas.org/cgi/doi/10.1073/pnas.1919344117

---

## Referee Comment (RC1) · Mark Blitz (Referee) · 19 Jun 2020

**Kinetics of dimethyl sulfide (DMS) reactions with isoprene-derived Criegee intermediates studied with direct UV absorption**

**General comments**

Time-resolved experiments have been carried out to generate two Criegee: CH2OO, which has been study many times; and MVKOO, which has only very recently been studied. In the presence of dimethyl sulphide, no additional Criegee removal was evident. Hence, only an upper limit is assigned for the rate coefficients. A theoretical potential energy surface has been calculated for $CH_2OO + (CH_3)_2S$ that has a significant barrier to products (DMSO + CH2O) and its rate coefficient is lower than the experimental upper limit. These results are clear-cut and only a few specific comments are raised.

If this were the only study on the titled reaction, the lack of reactivity would probably mean this paper would not be considered for publication in ACP. The reason this result is significant is that a previous study (Newland 2015) suggested the stabilized Criegee formed from O3/isoprene (mainly CH2OO/MKVO) react rapidly with dimethyl sulphide, with a rate coefficient close to the gas-kinetic frequency. As this other study generated the Criegees via ozonolysis (O3/isoprene), it does ask the question how we best understand ozonolysis in the atmosphere. Is stabilized Criegee chemistry the most important component of ozonolysis? More detail would help this paper. The comment from Andrew Rickard expands on this.

**Specific comments**

**Line 39** "*Surprisingly, the obtained rate coefficients are up to 104 times larger than previous results deduced from ozonolysis 40 experiments, indicating that the ozonolysis experiments could be quite complicated such that reliable kinetic results may be hard to retrieve.*"
This needs a reference. This is interesting in that relative rate experiments appear to be out by orders of magnitude. Is there explanation of these studies with today's knowledge? Is it wrong rate coefficients or is it more to do with the experiment itself?

**Line 103** "*To compensate for this effect, which was caused by the optics and the photolysis laser pulse, we recorded background traces without adding the precursor before and after each set of experiments. The reported data are after background subtraction.*" Can you state the typically size of this signal, i.e. what is I/I0 in the absence of added chemicals. Is it related to a heating effect?

**Line 134** "*e.g., bimolecular reactions with radical byproducts like I atoms, wall loss, etc.*" Probably self-reaction is most important. Any evidence for a second-order component?

This paper has probably done most to unravel the removal the kinetics in absence of added reagent.

CH2OO Criegee intermediate UV absorption cross-sections and kinetics of CH2OO + CH2OO and CH2OO + I as a function of pressureBy:Mir, ZS (Mir, Zara S.)[1]; Lewis, TR (Lewis, Thomas R.)[1]; Onel, L (Onel, Lavinia)[1]; Blitz, MA (Blitz, Mark A.)[1,2]; Seakins, PW (Seakins, Paul W.)[1]; Stone, D (Stone, Daniel)[1]

**Line 155** "*and show the results in Table S4.*", From Table S4, the results given in Figure 2 are fairly obvious. I would expected a similar result even if 248 nm photolysis was used. Significant photolysis of DMS could potentially lead to enhanced reactivity, and an energy dependence would be good practice. However, in the present case, there is no evidence of enhanced Criegee removal so there is not too much to worry about.

**Line 171** "*See SI (Sect. S3, page S5) for details.*" From the SI, the instant yield of MVKOO decreases with total pressure, which is consistent with population into CH3(C2H3)CIOO, i.e. the SV is linear. However, kr appears to be faster at low pressures. kr is the rate coefficient for the peroxy radical to react to MVKOO + I. It is not possible for a rate coefficient to increase at lower total pressure. There are too few pressures to say anything for definite, but it does highlight that the kr errors are not realistic.

I wonder if there is another explanation for the results in Table S3. If you had an additional species, produced from the photolysis of the di-iodo compound, X, that can react with the di-iodo compound to make the iodo radical.

di-iodo + hv ➔ X
X + di-iodo ➔ iodo radical

The pressure dependence could be linked to the fact the MVKOO species has a double bond.

If *k*r is the unimolecular reaction CH3(C2H3)CIOO ➔ MVKOO + I, then changing the temperature should be the easiest way to identify it.

**Line 172** "*This difference is consistent with the fact that MVKO is resonance-stabilized due to the extended conjugation of its vinyl group (Barber et al., 2018) and thus the adduct CH3(C2H3)CIOO is relatively less stable due to disruption of the conjugation.*"
It will be the properties of *CH3(C2H3)CIOO* that will most strongly influence its formation and unimolecular dissociation, *k*r.

**Line 194** *"Here we choose the boundary of three standard deviations as the upper limits for $kDMS+CI$, $kDMS+CH2OO \leq 4.2 \times 10 \cdot 15\ cm3s \cdot 1$ and $kDMS+MVKO \leq 1.6 \times 10 \cdot 14\ cm3s \cdot 1$"* As you have done calculations, it would be better to state that the expts provide only an upper limit, and it is most likely that the *k* are smaller and closer to the theoretical values.

**Line 203** *"[CI]ss is expected to be low, at least a couple of orders of magnitude lower than the steady-state [OH]ss."* On this basis, reactions need to be two orders of magnitude faster than OH to compete. $SO_2$, $H_2O$ vapour and acids fit the bill but not many other reagents.

**Line 216** *"While our direct measurements and kinetics are very straightforward, the ozonolysis experiments of Newland et al. might have been more complex than the authors (Newland et al., 2015) had assumed. For example, one may consider the possibility of converting DMS to SO2 via surface or gas-phase reactions (Chen et al., 2018) under the complicated conditions of isoprene ozonolysis."*
Is this a reasonable conclusion? In the introduction, you mentioned that prior to direct time-resolved experiments, Criegee + SO2 rate coefficients were thought to be slow. Is this another example of "surface" reactions? Is there more going in these ozonolysis experiments that bring about chemical change that if not captured by these direct measurements. Or are these relative rate reactions simply flawed?

Are there any suggested DMS ➜ SO2 schemes via the gas-phase?

**Line 220** Any reason why MKVOO + SO2 not calculated?

**TYPOS / UNDERSTANDING**

Is it MKVO or MVKOO? I think the later. This occurs several times

Also, MACRO or MACROO?

**Line 74** *"ozonlolysis"* Typo

**Line 91** *"However, DMS absorbs weakly at 248 nm. We therefore performed additional experiments by photolyzing CH2I2 at 248 nm to assess the impact of DMS photolysis at 248 nm on the decay of the CIs."* Do you mean 308 nm?

**Line 63** *"Newland et al. noted, however, that the presented rate coefficients do not correspond to the rates of single elementary reactions but rather describe the general reactivity of CIs towards DMS or H2O"* Can you re-phrase this as I'm not sure the point you making, be more explicit.

**Line 36** Beames et al., 2013 This is a depletion experiment.

---

## Referee Comment (RC2) · Anonymous Referee #2 · 12 Jul 2020

Title: Kinetics of dimethyl sulfide (DMS) reactions with isoprene-derived Criegee intermediates studied with direct UV absorption
Author(s): Mei-Tsan Kuo et al.
MS No.: acp-2020-484
MS Type: Research article

General Comments

The authors address an important result from the 2015 *ACP* paper of Newland *et al*.: dimethyl sulfide (DMS) reacts with Criegee intermediates (CIs) a factor of ~3 faster than $SO_2$ does with CIs; the CI + DMS rate constant is thus ~1 x $10^{-10}$ $cm^3$ $s^{-1}$. Based on their time-dependent measurements of CI concentration with UV absorption, the authors estimate upper limits for the CI + DMS rate constant of ~$10^{-15}$ $cm^3$ $s^{-1}$ for $CH_2OO$ and ~$10^{-14}$ $cm^3$ $s^{-1}$ for methyl vinyl ketone oxide (MVKO). Transition state theory (TST) calculations based on CCSD(T)//M06-2X quantum chemical data estimate the rate constant to be ~6 x $10^{-19}$ $cm^3$ $s^{-1}$.

The quality of the experimental and theoretical work is high and the results should be reliable. Where the manuscript could be strengthened is in its discussion of the previous work by Newland 2015. In this, I agree with the comments from Rickard and from Blitz. There should also be a slightly broader computational treatment of the relevant reactions along with a more detailed representation of the computational results.

Specific Comments

- The presentation of the theoretical predictions in Section 3.5 and Figure 5 would be much easier to follow if there were some graphical representation of the chemical structures being discussed.
- One reason for the difference is the current results and the results reported in Newland 2015 may be the impact of DMS on the MVKO + $SO_2$ reaction. It is not necessary to perform calculations on this reaction, but some mechanistic discussion would be pertinent.
- It could be enlightening to provide a brief context by comparing the title reaction with the previous study of $CH_3SH + CH_2OO$ (*J. Phys. Chem. A* **2019**, *123*, 4096-4103).
- **Line 224**: What is the evidence for the $CH_2OO$-DMS adduct having "very strong zwitterionic character?"
- **Supplemental Information S20-S21**: The authors should present some calculations on the MVKO. In particular, it would be worthwhile to consider how DMS might affect the cyclization of the *anti* conformer of MVKO to the dioxole (see *J. Am. Chem. Soc.* **2018**, *140*, 10866). Here, I reiterate the comment of Rickard that it would be useful for the authors to estimate the relative amounts of the *syn* and *anti* conformers of MVKO.

Technical Corrections

- **Lines 232-233**: "We did not examine more exotic CI reaction such as insertion in the DMS C–H bonds, as these are known to have comparatively high barriers." This statement should have a reference.

- **Supplemental Information S20-S21**:The authors should tabulate the relative energies predicted by the M06-2X/cc-pVDZ calculations.

---

## Referee Comment (RC3) · Anonymous Referee #3 · 30 Jul 2020

Kuo et al. report direct experimental and theoretical investigations of the reactions of two isoprene-derived Criegee intermediates with dimethyl sulfide (DMS). Using the diiodoalkane/diiodoalkene photolysis method to selectively generate each Criegee intermediate in turn, the authors probe the kinetics by UV absorption and deduce upper limit rate coefficients that are orders of magnitude slower than those obtained in the ozonolysis work of Newland et al. using the relative rate technique. The slow rate coefficient measured in the present work for $CH_2OO$ + DMS is substantiated by stationary point calculations coupled with CTST that yield a rate coefficient of $5.5E-19$ $cm^{-3}$ $s^{-1}$ at 298 K.

The paper is reasonably thorough and raises interesting discussion about ozonolysis vs. direct Criegee intermediate experimental kinetic studies, that have been significantly expanded by the other reviewers. The paper would benefit from some points of clarification (suggested below) and additional theoretical work on the MVK-oxide + DMS reaction to compare with the experimental results and contrast with the calculations on the $CH_2OO$ system. Please note that many of the comments in this review reflect the points that have already been raised in the thorough reviews of Rickard. Newland and Bloss, Blitz and the anonymous reviewer.

Main text

Page 2, line 33: The Welz et al. 2012 work is preceded by the Taatjes et al. JACS paper in which DMSO was used to generate the $CH_2OO$ Criegee intermediate.

Page 2, line 41: It is already established that ozonolysis experiments are by their very nature complicated – the authors should instead be more specific about the potential concerns they have regarding obtaining rate coefficients of Criegee intermediates from ozonolysis studies.

Page 2, line 51: The very recent Cox et al. paper in ACPD (https://www.atmos-chem-phys-discuss.net/acp-2020-472/) is also a thorough and up-to-date reference for existing studies of Criegee intermediate kinetics.

Page 3, line 90: A reference (or some further explanation) is needed regarding the MVKO precursor absorption at 308 nm.

Page 4, line 110: The authors should be able to determine an approximation of at least the MVK-oxide precursor concentration in their system. The vapor pressure of the precursor can be estimated using the Antoine coefficients. If the precursor was delivered to the reactor via a bubbler at a known flow rate, then the approximate concentration of the precursor can be deduced. In the event that the absorption coefficient of the precursor is deduced at a later date, this information would enable the concentration of

MVK-oxide used in the present work to be obtained.

Page 6, line 159: Under the present experimental conditions, CH3 would most likely undergo reaction with O2 to form CH3OO and so it would be best to compare the reactivity of CH3OO (rather than CH3) with I atom and Criegee intermediates.

Page 7, line 205: As the authors point out, there is currently significant uncertainty in the estimated and modelled steady state concentrations of Criegee intermediates. Because of this, it would be instructive to also frame the competitiveness of Criegee-initiated DMS oxidation vs. OH or NO3-initiated oxidation in terms of what concentration of Criegee intermediates are needed to oxidize a certain fraction (e.g. 5%, 10% or 20%) of atmospheric DMS using the theoretically determined rate coefficient.

Page 8, line 220. It seems peculiar that you have chosen to investigate theoretically only the CH2OO reaction and not the MVK-oxide reaction also. In MVK-oxide, the conjugation of the unsaturated side chain with the carbonyl oxide group has the potential to substantially alter the surface. These calculations are likely significantly more complex than for the CH2OO case because of the need to consider syn and anti conformers, and cis/trans forms of each of these. However, given the interesting structural and conformeric dependence of Criegee intermediate reactivity, it is a regretful omission.

Page 8, line 233: A reference is needed to substantiate the statement regarding high barriers for DMS C-H insertion.

Page 8, line 251: Do you anticipate stabilization of the (CH3)2SCH2OO adduct under tropospheric conditions?

Page 9, line 261: You hypothesize that surface reactions converting DMS to SO2 in the chamber study of Newland could be the source of discrepancy between the present work and the work of Newland et al. I encourage the authors to respond to the comments of Rickard, Newland and Bloss, and Blitz regarding this matter.

Figure 2: Please include a note to address if the error bars are included or not included

on this plot (as noted for Figure 4). Given that the rate coefficients for the self-reaction of CH2OO and the reaction of CH2OO + I (see Blitz review) are now well established, it would be pertinent to deduce which of these is the major source of increased loss rates at higher laser fluence are under your experimental conditions.

Supplementary information

Table S3: Because both the reaction forming MVK-oxide from the precursor + O2 reaction as well as the MVK-oxide + SO2 reaction features an adduct, the authors should label more caerefully the adduct referred to in the 'adduct yield' column of the table to avoid confusion.

Figures S1, S2: Provide details about error bars (c.f. comment about Figure 2).

Figure S4: Please discuss the proposed origin of the "spike" at time zero.

S10: These additional investigations are illuminating and interesting.

Additional comments regarding MVK-oxide conformers

I would like to add some discussion to the comments made by other reviewers regarding which conformers of MVK-oxide are produced from the photolytic scheme vs. ozonolysis. While the distribution of these conformers has not yet been deduced, the recent literature on direct MVK-oxide kinetic and spectroscopic studies that indicated that both syn (Caravan et al., PNAS 2020) and anti (Vansco et al., JPCA 2020) confirmers are produced from the 1,3,-diiodobut-2-ene photolysis scheme used in the present work (Barber et al., JACS 2018). Additionally, due to the rapid unimolecular decay of anti compared with syn (Barber et al., JACS 2018 and Vereecken et al., PCCP 2017), it is unlikely that reaction with DMS could compete with unimolecular decay under tropospheric conditions for the anti conformer.

---

## Author Comment (AC1) · 27 Aug 2020

We thank the referees for their careful reading of the manuscript and helpful comments, which are repeated below (in black font). Our replies are given in blue font directly after each comment.

**Referee 1:**

**General comments**

Time-resolved experiments have been carried out to generate two Criegee: $CH_2OO$, which has been study many times; and MVKOO, which has only very recently been studied. In the presence of dimethyl sulphide, no additional Criegee removal was evident. Hence, only an upper limit is assigned for the rate coefficients. A theoretical potential energy surface has been calculated for $CH2OO + (CH3)2S$ that has a significant barrier to products (DMSO + $CH_2O$) and its rate coefficient is lower than the experimental upper limit. These results are clear-cut and only a few specific comments are raised.

If this were the only study on the titled reaction, the lack of reactivity would probably mean this paper would not be considered for publication in ACP. The reason this result is significant is that a previous study (Newland 2015) suggested the stabilized Criegee formed from $O_3$/isoprene (mainly $CH_2OO$/MKVO) react rapidly with dimethyl sulphide, with a rate coefficient close to the gas-kinetic frequency. As this other study generated the Criegees via ozonolysis ($O_3$/isoprene), it does ask the question how we best understand ozonolysis in the atmosphere. Is stabilized Criegee chemistry the most important component of ozonolysis? More detail would help this paper. The comment from Andrew Rickard expands on this.

**Specific comments**

**Line 39** "*Surprisingly, the obtained rate coefficients are up to $10^4$ times larger than previous results deduced from ozonolysis experiments, indicating that the ozonolysis experiments could be quite complicated such that reliable kinetic results may be hard to retrieve.*"

This needs a reference. This is interesting in that relative rate experiments appear to be out by orders of magnitude. Is there explanation of these studies with today's knowledge? Is it wrong rate coefficients or is it more to do with the experiment itself?

AUTHORS' REPLY:

Welz et al. (2012) compared their rate coefficients with previous values applied in contemporary tropospheric models (Johnson and Marston, 2008; Johnson et al., 2001; Hatakeyama and Akimoto, 1994). For ozonolysis experiments, typically only the ratios of reaction rate coefficients, e.g. $k_{DMS}/k_{SO2}$ (Newland et al., 2015), are obtained. The researchers have to compare with (at least) one absolute rate coefficient to get the rest rate coefficients. Unfortunately, the selected absolute rate coefficient (at that time) has large uncertainty, which propagates to other reported values. In addition, the reaction mechanism may be rather complicated and even the ratios of the rate coefficients must be treated with care. The above three references will be included in the main text.

**Line 103** "*To compensate for this effect, which was caused by the optics and the photolysis laser pulse, we recorded background traces without adding the precursor before and after each set of experiments. The reported data are after background subtraction.*" Can you state the typically size of this signal, i.e. what is I/I0 in the absence of added chemicals. Is it related to a heating effect?

AUTHORS' REPLY:

Typical background traces as well as raw signal traces (without background subtraction) obtained at 248 nm and 308 nm will be shown in Figures S5 and S6, respectively. These backgrounds are originated from the different longpass filters used for coupling the laser beam and probe beam into the reactor. Yes, it is likely that the backgrounds are from a heating effect of the longpass filters.

[Figure]

Fig. S5. Background traces under normal DMS concentrations, represented in colour lines, and the raw signal traces (without background subtraction), represented in grey lines, obtained with 248 nm photolysis laser ($I_{248nm}$ = 2.43 mJ cm$^{-2}$). See Exp#22 of Table S3 for the experimental conditions.

[Figure]

Fig. S6. Background traces under normal DMS concentrations, represented in colour lines, and the raw signal traces (without background subtraction), represented in grey lines, obtained with 308 nm photolysis laser ($I_{308nm}$ = 2.35 mJ cm$^{-2}$). See Exp#2 of Table S1 for the experimental condition. Note that the optics (longpass filters) are different from those at 248 nm.

**Line 134** "*e.g., bimolecular reactions with radical byproducts like I atoms, wall loss, etc.*" Probably self-reaction is most important. Any evidence for a second-order component?

This paper has probably done most to unravel the removal the kinetics in absence of added

reagent. CH2OO Criegee intermediate UV absorption cross-sections and kinetics of CH2OO + CH2OO and CH2OO + I as a function of pressure By:Mir, ZS (Mir, Zara S.)[ 1 ] ; Lewis, TR (Lewis, Thomas R.)[ 1 ] ; Onel, L (Onel, Lavinia)[ 1 ] ; Blitz, MA (Blitz, Mark A.)[ 1,2 ] ; Seakins, PW (Seakins, Paul W.)[ 1 ] ; Stone, D (Stone, Daniel)[ 1 ]

AUTHORS' REPLY:

In the previous works of Smith et al. (Smith et al., 2016) and Li et al. (Li et al., 2020), the contributions of the pseudo-first-order reactions and second-order reactions are both considered and the kinetic model can be represented in the following equation:

$$\frac{-d[\text{CI}]}{dt} = k_1[\text{CI}] + k_2[\text{CI}]^2$$

The above equation can be simplified when extrapolating the rate coefficients to zero concentration of $[\text{CI}]_0$:

$$\frac{-d[\text{CI}]}{dt} \cong (k_1 + \frac{1}{2}k_2[\text{CI}]_0)[\text{CI}] = k_{\text{obs}}[\text{CI}]$$

The difference between the complete and simplified equations only shows up at high $[\text{CI}]_0$. Most important of all, the self-reaction of CIs would not affect the determination of $k_{\text{DMS}}$, since $[\text{CI}]_0$ was kept constant in every experimental set.

Based on the absolute absorption cross section of $\text{CH}_2\text{OO}$ at 340 nm ($\sigma = 1.23 \times 10^{-17}$ cm$^2$) and the pressure-dependent yield of $\text{CH}_2\text{OO}$ from $\text{CH}_2\text{I} + \text{O}_2$ (0.46 at 300 Torr) (Ting et al., 2014a) the number densities of relevant species can be estimated to be the following (for Exp#1, Table S1).

$[\text{CH}_2\text{OO}]_0 = 6.7 \times 10^{11}$ cm$^{-3}$; $[\text{I}]_0 = 2.1 \times 10^{12}$ cm$^{-3}$; $[\text{CH}_2\text{IOO}]_0 = 7.7 \times 10^{11}$ cm$^{-3}$.

The first-order decay rate coefficient of $\text{CH}_2\text{OO}$ ($k_{\text{eff}}$) can be approximately estimated (Li et al., 2020) as:

$$k_{\text{eff}} = k_{\text{I}}[\text{I}]_0 + k_{\text{self}}[\text{CH}_2\text{OO}]_0$$

Using $k_{\text{self}} = 8 \times 10^{-11}$ cm$^3$ s$^{-1}$ and $k_{\text{I}} = 5.8 \times 10^{-11}$ cm$^3$ s$^{-1}$ at 300 Torr (Mir et al., 2020), the estimated $k_{\text{eff}}$ is 180 s$^{-1}$, consistent with the observed value of 232 s$^{-1}$ for $k_0$. Therefore, the main loss processes of $\text{CH}_2\text{OO}$ are reaction with iodine atoms (and other radicals) and its self-reaction.

In Figure S7, we can see a nice linear relationship between $k_0$ and the total produced radicals (proportional to the product of the laser fluence and the precursor concentration), further supporting the above mechanism. We would add the following sentences in the caption of Figure S7.

*"The main loss processes of CH$_2$OO are reactions with radical byproducts like iodine atoms and its self-reaction. The observed values of $k_0$ (e.g., 232 s$^{-1}$ for Exp#1) are consistent with the values (e.g., 180 s$^{-1}$ at the condition of Exp#1) that are estimated using the reported kinetic data (yield and rate coefficients) (Mir et al., 2020; Ting et al., 2014)."*

We will also modify the relevant sentences in the main text to:

*"The subsequent decay in absorption is due to the consumption of CH$_2$OO either through reaction with DMS or through other processes, e.g., bimolecular reactions with radical byproducts like I atoms, wall loss, etc. In addition, self-reaction of CH$_2$OO has been found to be rather fast ($k_{self} = 8 \times 10^{-11} cm^3 s^{-1}$)(Mir et al., 2020). However, the effect of the self-reaction (Smith et al., 2016;Li et al., 2020) would not affect the determination of $k_{DMS}$ under our experimental conditions."*

**Line 155** "*and show the results in Table S4.*" , From Table S4, the results given in Figure 2 are fairly obvious. I would expected a similar result even if 248 nm photolysis was used. Significant photolysis of DMS could potentially lead to enhanced reactivity, and an energy dependence would be good practice. However, in the present case, there is no evidence of enhanced Criegee removal so there is not too much to worry about.

AUTHORS' REPLY:

In Table S4, we have shown that [DMS]$_{diss}$ is about ten times less than [CH$_2$I$_2$]$_{diss}$ under typical experimental conditions when 248 nm photolysis is applied. However, we have observed a strong absorption in the background traces when 248 nm photolysis and high [DMS] are applied (Figure S4). The extra absorption from the dissociated DMS would be problematic when performing the background subtraction. Therefore we constrained the laser fluence and [DMS] to preclude the influence of [DMS] photolysis.

**Line 171** "*See SI (Sect. S3, page S5) for details.*" From the SI (Table S3), the instant yield of MVKOO decreases with total pressure, which is consistent with population into $CH_3(C_2H_3)CIOO$, i.e. the SV is linear. However, $k_r$ appears to be faster at low pressures. $k_r$ is the rate coefficient for the peroxy radical to react to MVKOO + I. It is not possible for a rate coefficient to increase at lower total pressure. There are too few pressures to say anything for definite, but it does highlight that the $k_r$ errors are not realistic.

I wonder if there is another explanation for the results in Table S3. If you had an additional species, produced from the photolysis of the di-iodo compound, X, that can react with the di-iodo compound to make the iodo radical.

$$di\text{-}iodo + h\nu \rightarrow X$$

$$X + di\text{-}iodo \rightarrow iodo\ radical$$

The pressure dependence could be linked to the fact the MVKOO species has a double bond.

If $k_r$ is the unimolecular reaction $CH_3(C_2H_3)CIOO \rightarrow MVKOO + I$, then changing the temperature should be the easiest way to identify it.

**Line 172** "*This difference is consistent with the fact that MVKO is resonance-stabilized due to the extended conjugation of its vinyl group (Barber et al., 2018) and thus the adduct $CH_3(C_2H_3)CIOO$ is relatively less stable due to disruption of the conjugation.*"

It will be the properties of *$CH_3(C_2H_3)CIOO$* that will most strongly influence its formation and unimolecular dissociation, $k_r$.

AUTHORS' REPLY (To lines 171-172):

The reviewer is right about the role of $CH_3(C_2H_3)CIOO$ and that changing the temperature should be the easiest way to identify the related process. In fact, we have discussed the issues of the adduct, including the temperature and pressure effects, in our recent paper (Lin et al., 2020). Since MVKO is a resonance-stabilized molecule, adduct would be relatively less stable, compared with CIs without resonance structure, such as $CH_2OO$ or $CH_3CHOO$. Therefore, the unimolecular decomposition of the adduct is observed in our experimental time scale. The reason why $k_r$ appears to be larger at lower pressure is that the fitted $k_r$ should include the unimolecular decomposition of the adduct and the reaction of the adduct with other radicals (X) such as iodine atoms.

$$k_r = k_{uni}(\text{adduct}) + k_x[\text{X}]$$

The concentration of the radicals would increase as the precursor concentration increases, leading to a higher $k_r$. This relation can be observed explicitly through plotting $k_r$ against $I_{248}$ $_{nm} \times Abs$(238 nm) (photolysis laser fluence times precursor absorbance). As for the temperature effect, we have also observed a positive temperature dependence of $k_r$ ($E_a =$ 12.7±0.3 kcal mol$^{-1}$), consistent with the calculation result for the bond dissociation energy of the adduct (14 kcal mol$^{-1}$) (Lin et al., 2020).

[Figure]

Fig. S?. Plot of $k_r$ against the product of the laser fluence ($I_{248nm}$) and the absorbance of 1,3-diiodo-2-butene at 238 nm in the photolysis cell ($Abs$(238nm)) for the experiments of MVKO+DMS reaction (Exp#15-29, Tables S3). The x-axis essentially represents the total amounts of radical species generated through the photolysis of the precursor (R1) and the subsequent reactions (R2). Higher radical concentration results in faster decay of the adduct, thus higher $k_r$.

Please note that the error bars in Tables S1-S3 do NOT include any systematic errors. For $k_r$, it is correlated with other fitting parameters like (1−$\alpha$). Since MVKO does not react with DMS (essentially all the traces are almost the same at various [DMS]), it is hard to 'disentangle' the correlation among fitting parameters. In the paper by Lin et al., we used SO$_2$ to scavenge MVKO and to obtain more robust results (Lin et al., 2020).

We will add a notation regarding the error bar of $k_r$ after Table S3:

"*averaged value ± 1 sigma error of the mean (statistical only, not including systematic*

*errors). The actual error bar would be larger since $k_r$ is highly correlated with other fitting parameters like (1−α). Lin et al. has used $SO_2$ scavenger to obtain more robust results for $k_r$ (Lin et al., 2020)."*

**Line 194** "*Here we choose the boundary of three standard deviations as the upper limits for $k_{DMS}+CI$, $k_{DMS}+CH_2OO \leq 4.2 \times 10^{15}$ cm$^3$s$^{-1}$ and $k_{DMS}+MVKO \leq 1.6 \times 10^{14}$ cm$^3$s$^{-1}$*" As you have done calculations, it would be better to state that the expts provide only an upper limit, and it is most likely that the $k$ are smaller and closer to the theoretical values.

AUTHORS' REPLY

Indeed, the actual value of $k_{DMS+CI}$ would be smaller than the upper limits we reported, and the actual value of $k_{DMS+CH2OO}$ may be closer to the theoretical value ($k_{DMS+CH2OO} = 5.5 \times 10^{-19}$ cm$^3$ s$^{-1}$). However, the calculation is not at the best level (while it is still good enough for the discussion in this paper) and there are uncertainties in the calculated values. Thus we decided not to say that the rate coefficients would be closer to the theoretical values.

**Line 203** "*[CI]ss is expected to be low, at least a couple of orders of magnitude lower than the steady-state [OH]ss.*" On this basis, reactions need to be two orders of magnitude faster than OH to compete. SO2, H2O vapour and acids fit the bill but not many other reagents.

AUTHORS' REPLY

We totally agree with your point. Thus we think the reaction of CI+DMS would not be a major path for the oxidization of DMS since the rate coefficient of CI+DMS is quite small.

**Line 216** "*While our direct measurements and kinetics are very straightforward, the ozonolysis experiments of Newland et al. might have been more complex than the authors (Newland et al., 2015) had assumed. For example, one may consider the possibility of converting DMS to SO2 via surface or gas-phase reactions (Chen et al., 2018) under the complicated conditions of isoprene ozonolysis.*"

Is this a reasonable conclusion? In the introduction, you mentioned that prior to direct timeresolved experiments, Criegee + SO$_2$ rate coefficients were thought to be slow. Is this another example of "surface" reactions? Is there more going in these ozonolysis experiments that bring about chemical change that if not captured by these direct measurements. Or are these relative rate reactions simply flawed?

Are there any suggested DMS $\rightarrow$ SO$_2$ schemes via the gas-phase?

AUTHORS' REPLY

The reviewer raised a few important and interesting questions, which are awaiting more investigations. As mentioned before, researchers have to postulate the reaction mechanism of the ozonolysis reaction to deduce the rate coefficients. We believe there are more to be studied for the ozonolysis of isoprene. For clarification, we would modify the related text to the following.

*"For the determination of the relative rate of the CI + DMS reaction, Newland et al. monitored the consumption of SO$_2$ over a measurement period of up to 60 min until approximately 25% of isoprene was consumed (Newland et al., 2015). Additional uncharacterized reaction pathways (e.g., reactions with the products) would lead to a bias in the inferred rate coefficients. A part of this high complexity of the isoprene-ozone-DMS-SO$_2$ system has been discussed by Newland et al. in the section of Experimental Uncertainties (Newland et al., 2015). Our direct measurements and kinetics are very straightforward; the obtained results for individual CIs may provide useful constraints for related ozonolysis systems."*

**Line 220** Any reason why MKVOO + SO$_2$ not calculated?

AUTHORS' REPLY

We guess the reviewer meant MVKO+DMS. Now we have the calculation result of MVKO+DMS reaction. Similar to the reaction with H$_2$O (Vereecken et al., 2017 ), the direct reaction of *E*- and *Z*-MVKO with DMS is expected to be slower than for CH$_2$OO, as the organic groups and the conjugation of the carbonyl oxide moiety with the double bond stabilizes the CI. Indeed, for MVKO (all conformers), no adduct seems to exist at the M06-2X/cc-pVDZ level of theory: the needed C$-$S bond in the adduct appears to be too weak to

compensate for the loss of the conjugation in the carbonyl oxide, and the system reverts to the MVKO + DMS complex instead, without a formal C−S bond. As a result, the barrier for the migration of a DMS methyl H-atom to the carbonyl oxide oxygen to form a methylidene adduct is ~10 kcal/mol higher than for the analogous TS in the $CH_2OO$+DMS system which does feature a weakly bonded intermediate adduct. The direct oxygen transfer from *E*- or *Z*-MVKO to DMS, forming MVK + DMSO, was found to have a similarly high energy barrier as in the $CH_2OO$+DMS system. No viable reaction channels were found involving the double bond in MVKO. The lack of accessible transition states then prohibits rapid direct reaction between DMS and MVKO.

We also have additional calculation on the cyclisation of MVKO in the presence of DMS. Again, no accessible pathways were found.

**TYPOS / UNDERSTANDING**

Is it MKVO or MVKOO? I think the later. This occurs several times

Also, MACRO or MACROO?

AUTHORS' REPLY

MVKO is short for methyl-vinyl-ketone-oxide, and is the correct notation (i.e. MVK + 1 oxide O-atom). Likewise, MACRO is an acronym for methacroleine-oxide. We have standardized on these notations, consistent with our previous paper (Lin et al., 2020).

**Line 74** "ozonlolysis" Typo

AUTHORS' REPLY

(will be fixed). Thanks for your reminder.

**Line 91** "However, DMS absorbs weakly at 248 nm. We therefore performed additional experiments by photolyzing CH2I2 at 248 nm to assess the impact of DMS photolysis at 248 nm on the decay of the CIs."

Do you mean 308 nm?

AUTHORS' REPLY

We want to emphasize that DMS absorbs weakly at 248 nm ($\sigma = 1.28\times10^{-20}$ cm$^2$) but barely absorbs at 308 nm ($\sigma < 1\times10^{-22}$ cm$^2$) (Limão-Vieira et al., 2002). At low [DMS], the weak absorption of DMS at 248 nm may not cause a problem, but in this work, [DMS] is quite high and thus the photolysis of DMS at 248 nm should be taken into consideration.

**Line 63** "*Newland et al. noted, however, that the presented rate coefficients do not correspond to the rates of single elementary reactions but rather describe the general reactivity of CIs towards DMS or H2O*" Can you re-phrase this as I'm not sure the point you making, be more explicit.

AUTHORS' REPLY:

Thank you for pointing out. The sentences will be rephrased to

"*Newland et al., who used ozonolysis of isoprene to generate a mixture of CIs (CH$_2$OO, MVKO, and MACRO), reported a combined reactivity of these CIs toward DMS and H$_2$O under conditions similar to the atmospheric boundary layer. Their reported rate coefficients might not correspond to those of single elementary reactions.*"

**Line 36** Beames et al., 2013 This is a depletion experiment.

AUTHORS' REPLY:

Thank you for pointing out. The sentence will be rephrased to

"… UV-visible absorption/depletion spectroscopy …"

**Anonymous referee 2:**

One reason for the difference is the current results and the results reported in Newland 2015 may be the impact of DMS on the MVKO + $SO_2$ reaction. It is not necessary to perform calculations on this reaction, but some mechanistic discussion would be pertinent.

AUTHORS' REPLY:

The reactions of carbonyl oxides (CI) with $SO_2$ proceed by a barrierless cycloaddition (Kuwata et al., J. Phys. Chem. A, 119, 10316, 2015) with a very fast capture rate coefficient for complex formation near the collision limit, and a partial redissociation to the free reactants leading to a rate coefficient somewhat below the collision limit. The DMS-complex of a CI reacting with $SO_2$ can be expected to have a lower rate coefficient than the direct CI+$SO_2$ reaction, as the DMS shields part of the approach vectors of the $SO_2$ reactant, and the long-range attractive force is diminished due to a somewhat lower dipole moment of the complex compared to the free CI. However, the reduction of the rate coefficient is not expected to be all that large, and more importantly the CI+DMS complex is not overly strong such that only a small fraction of the CI will be present as a CI+DMS complex. This makes it hard to understand how DMS could affect any CI+$SO_2$ capture reaction ($CH_2OO$, MVKO, or $CH_3CHOO$) to the extent observed in Newland et al. It is for this reason that we have done exploratory calculations on the redissociation of the CI+$SO_2$ cyclo-adduct, but have found no indication that this would have the required impact on the effective CI+DMS rate of product formation.

**Line 224**: What is the evidence for the $CH_2OO$-DMS adduct having "very strong zwitterionic character?"

AUTHORS' REPLY:

At the level of theory used here, the wavefunction for the adduct converges to a closed-shell structure with no biradical character, where the O-atoms have a strongly negative partial charge (up to -0.46 in the Mulliken population analysis), and where the S-atom is positively charged S-atom (+0.28 in the Mulliken population analysis, compared to the Mulliken partial charge of -0.06 in DMS). This suggests that the $CH_2OO$-DMS adduct, similar to the parent

carbonyl oxide, has a zwitterionic character with very strong charge separation between the S and O atoms, rather than a biradical wavefunction.

**Supplemental Information S20-S21**: The authors should present some calculations on the MVKO. In particular, it would be worthwhile to consider how DMS might affect the cyclization of the anti conformer of MVKO to the dioxole (see J. Am. Chem. Soc. 2018, 140, 10866). Here, I reiterate the comment of Rickard that it would be useful for the authors to estimate the relative amounts of the syn and anti conformers of MVKO.

AUTHORS' REPLY:

The dominant unimolecular reaction of $E$-MVKO is a 1,4-H-shift (VHP-channel), analogous to $Z$-CH$_3$CHOO, for which we already showed that any catalytic effect is insufficient to allow for fast reactions. We now also calculated the impact of a DMS spectator complexing agent on the cyclization in $Z$-MVKO at the M06-2X/cc-pVDZ level of theory, finding similar results as for the methylated CH$_3$CHOO, i.e. the barrier height without (12.1 kcal/mol) and with complexing DMS (14.2 kcal/mol from the ground state of the complex) are essentially identical. The complex stability for $Z$-MVKO + DMS ($-9.9$ kcal/mol) is also similar to that for CH$_2$OO, $Z$-CH$_3$CHOO, and $E$-CH$_3$CHOO. Any catalyzing effect by DMS would then be due to chemical activation by the energy released in the complexation. The net energy barrier for the DMS catalysed $Z$-MVKO unimolecular reaction is ~ +4 kcal/mol, then still implies a slow bimolecular reaction, in agreement with the experimental observations.

Also see Reply to Referee 1 (for Line 220) for the calculation results on the direct reaction of MVKO + DMS.

Regarding the relative amounts of the *syn* and *anti* conformers of MVKO, we would add the following sentences to clarify the MVKO conformation. (after line 80)

*"For MVKO, there are 4 possible conformers. Following the nomenclature of Barber et al., syn/anti-MVKO (E/Z-MVKO) has a methyl/vinyl group at the same side of the terminal oxygen, while cis and trans refer to the orientation between the vinyl C=C and the carbonyl C=O bonds (Barber et al., 2018). It has been reported that syn- and anti-MVKO do not interconvert due to a high barrier between them but the barrier between cis and trans forms is low enough to permit fast interconversion at 298 K (Barber et al., 2018;Vereecken et al., 2017). Caravan et al., have shown that anti-MVKO is unobservable under thermal (298 K) conditions due to short lifetime and/or low yield, and thus, the UV-Vis absorption signal is from an equilibrium mixture of cis and trans forms of syn-MVKO (Caravan et al., 2020;Vereecken et al., 2017). For simplicity we will use MVKO to represent syn-MVKO (E-MVKO)."*

**Lines 232-233**: "*We did not examine more exotic CI reaction such as insertion in the DMS C–H bonds, as these are known to have comparatively high barriers.*" This statement should have a reference.

AUTHORS' REPLY:

We would add the paper of (Decker et al. ***Phys. Chem. Chem. Phys.***, 2017,**19**, 8541-8551, doi:10.1039/C6CP08602K) into the reference

**Supplemental Information S20-S21**: The authors should tabulate the relative energies predicted by the M06-2X/cc-pVDZ calculations.

AUTHORS' REPLY:

A table is now included in the supporting information.

Table S_: ZPE-corrected DMS complex energies, E(complex), and barrier heights $E_b$ without and with a DMS complexing agent, at the M06-2X/cc-pVDZ level of theory. Energies are in kcal mol$^{-1}$ and relative to the free reactants.

| CI reaction | $E_b$ | E(complex) | $E_b$(complex) |
|---|---|---|---|
| $CH_2OO \rightarrow cyc\text{-}CH_2OO\text{-}$ | 22.0 | -9.6 | 14.5 |
| $Z\text{-}CH_3CHOO \rightarrow CH_2CHOOH$ | 12.7 | -8.6 | 7.2 |
| $Z\text{-}CH_3CHOO \rightarrow cyc\text{-}CH(CH_3)OO\text{-}$ | 25.8 | -8.6 | 18.2 |
| $E\text{-}CH_3CHOO \rightarrow cyc\text{-}CH(CH_3)OO\text{-}$ | 18.4 | -10.9 | 9.5 |
| $Z\text{-}(CH=CH_2)C(CH_3)OO \rightarrow cyc\text{-}CH\text{-}CH_2C(CH_3)OO\text{-}$ | 12.1 | -9.9 | 4.4 |
| $Z\text{-}(CH=CH_2)C(CH_3)OO + DMS \rightarrow MVK + DMSO$ | 8.7 | | |
| $E\text{-}(CH_3)C(CH=CH_2)OO + DMS \rightarrow MVK + DMSO$ | 8.0 | | |
| $(CH_3)C(CH=CH_2)OO + DMS \rightarrow$ $S(CH_3)(=CH_2)C(CH_3)(CH=CH_2)OOH$ | 11.2 | | |

**Anonymous Referee #3**

Kuo et al. report direct experimental and theoretical investigations of the reactions of two isoprene-derived Criegee intermediates with dimethyl sulfide (DMS). Using the diiodoalkane/diiodoalkene photolysis method to selectively generate each Criegee intermediate in turn, the authors probe the kinetics by UV absorption and deduce upper limit rate coefficients that are orders of magnitude slower than those obtained in the ozonolysis work of Newland et al. using the relative rate technique. The slow rate coefficient measured in the present work for $CH_2OO$ + DMS is substantiated by stationary point calculations coupled with CTST that yield a rate coefficient of 5.5E-19 cm-3 s-1 at 298 K.

The paper is reasonably thorough and raises interesting discussion about ozonolysis vs. direct Criegee intermediate experimental kinetic studies, that have been significantly expanded by the other reviewers. The paper would benefit from some points of clarification (suggested below) and additional theoretical work on the MVK-oxide + DMS reaction to compare with the experimental results and contrast with the calculations on the $CH_2OO$ system. Please note that many of the comments in this review reflect the points that have already been raised in the thorough reviews of Rickard. Newland and Bloss, Blitz and the anonymous reviewer.

Main text

**Page 2, line 33**: The Welz et al. 2012 work is preceded by the Taatjes et al. JACS paper in which DMSO was used to generate the $CH_2OO$ Criegee intermediate.

AUTHORS' REPLY:

The reviewer is correct. However, the method reported by Taatjes et al.(Taatjes et al., 2008) is less efficient than that by Welz et al. (Welz et al., 2012) Nowadays, most photolytic generation of Criegee intermediates follow the method by Welz et al. The related sentences

*"However, due to their high reactivity and, hence, short lifetimes, laboratory studies of the reactions of CIs have been challenging. In fact, no direct detection of CIs has been known before Welz et al. reported a novel method to efficiently generate CIs other than through ozonolysis of alkenes (Welz et al., 2012)."*

would be modified to

*"However, due to their high reactivity and, hence, short lifetimes, laboratory studies of the reactions of CIs have been challenging until the work by Welz et al. who reported a novel*

*method to efficiently generate CIs other than through ozonolysis of alkenes (Welz et al., 2012)."*

**Page 2, line 41**: It is already established that ozonolysis experiments are by their very nature complicated – the authors should instead be more specific about the potential concerns they have regarding obtaining rate coefficients of Criegee intermediates from ozonolysis studies.

AUTHORS' REPLY:

We will clarify the situation by revising the related text to

*"Surprisingly, the obtained rate coefficients are up to $10^4$ times larger than previous results deduced from ozonolysis experiments (Johnson and Marston, 2008;Johnson et al., 2001;Hatakeyama and Akimoto, 1994). For ozonolysis experiments, typically only the ratios of reaction rate coefficients are obtained. The researchers have to compare with (at least) one absolute rate coefficient to get the rest rate coefficients. Unfortunately, the selected absolute rate coefficient (at that time) has large uncertainty, which propagates to other reported values. In addition, the reaction mechanism may be rather complicated and even the ratios of the rate coefficients need to be treated with care."*

**Page 2, line 51:** The very recent Cox et al. paper in ACPD (https://www.atmos-chemphys-discuss.net/acp-2020-472/) is also a thorough and up-to-date reference for existing studies of Criegee intermediate kinetics.

AUTHORS' REPLY:

Thanks. We will include this new reference. (Cox et al., 2020)

**Page 3, line 90:** A reference (or some further explanation) is needed regarding the MVKO precursor absorption at 308 nm.

AUTHORS' REPLY:

Below would be the measured absorbance of the diiodomethane (Exp# 12) and 1,3-diiodo-2-butene (Exp# 15) in the absorption cell (they are much diluted in the reactor cell). The absorption of 1,3-diiodo-2-butene at 308 nm is c.a. one-tenth of that at 248 nm. Consequently, we only perform the photolysis of 1,3-diiodo-2-butene at 248 nm.

[Figure]

**Page 4, line 110:** The authors should be able to determine an approximation of at least the MVK-oxide precursor concentration in their system. The vapor pressure of the precursor can be estimated using the Antoine coefficients. If the precursor was delivered to the reactor via a bubbler at a known flow rate, then the approximate concentration of the precursor can be deduced. In the event that the absorption coefficient of the precursor is deduced at a later date, this information would enable the concentration of MVK-oxide used in the present work to be obtained.

AUTHORS' REPLY:

Currently we don't have the available data for the cross section nor the empirical coefficients of Antoine coefficients for 1,3-diiodo-2-butene; hence we couldn't derive the absolute concentration. We have reported the deduced absorbance (*Abs*) of the precursor in the photolysis cell of different experiments sets in Table S3. The absolute concentration of precursor can be deduced from the *Abs* of precursor and other experimental conditions shown in Table S3., once the absolute cross section of 1,3-diiodo-2-butene is available.

We have modified the text to

*"However, because no absolute absorption cross sections for 1,3-diiodo-2-butene have been reported, its absolute concentration cannot be determined. We alternatively report the absorbance (Precursor Abs) of 1,3-diiodo-2-butene in the photolysis reactor (Table S3)."*

**Page 6, line 159:** Under the present experimental conditions, $CH_3$ would most likely undergo reaction with $O_2$ to form $CH_3OO$ and so it would be best to compare the reactivity of $CH_3OO$ (rather than $CH_3$) with I atom and Criegee intermediates.

AUTHORS' REPLY:

Thanks for pointing out. We will revise the sentences to

*"The expected products of DMS photolysis are $CH_3$ + $CH_3S$ (Bain et al., 2018). Under the presence of $O_2$ (10 Torr), $CH_3$ would be converted into $CH_3OO$. These radicals ($CH_3$, $CH_3OO$, and $CH_3S$) are less reactive than I atoms or CIs."*

**Page 7, line 205:** As the authors point out, there is currently significant uncertainty in the estimated and modelled steady state concentrations of Criegee intermediates. Because of this, it would be instructive to also frame the competitiveness of Criegee-initiated DMS oxidation vs. OH or NO3-initiated oxidation in terms of what concentration of Criegee intermediates are needed to oxidize a certain fraction (e.g. 5%, 10% or 20%) of atmospheric DMS using the theoretically determined rate coefficient.

AUTHORS' REPLY:

Possible concentrations of $NO_3$ and OH in the troposphere are found to be:

[OH] = $1 \times 10^6$ cm$^{-3}$ (Li et al., 2018) and [$NO_3$] = 10 ppt = $2.5 \times 10^8$ cm$^{-3}$ (Khan et al., 2015). Together with the reaction rate coefficients ($k_{DMS+OH}$ = $4.8 \times 10^{-12}$ cm$^3$ s$^{-1}$, $k_{DMS+NO3}$ = $6.8 \times 10^{-11}$ cm$^3$ s$^{-1}$ (Atkinson et al., 2004)), the concentration of CIs would have to be unreasonably high, at the order of $10^{11}$ cm$^{-3}$, to be competitive (5% of the effective reaction rate) with the DMS+OH and DMS+$NO_3$ reactions.

We would add the following sentences

*"If the DMS reactions with CIs were to be competitive to those with $NO_3$ (e.g., $2.5 \times 10^8$ cm$^{-3}$) and OH (e.g., $1 \times 10^6$ cm$^{-3}$) (e.g., 5% of the overall DMS removal), the concentration of CIs would have to be unreasonably high, at the order of $10^{11}$ cm$^{-3}$."*

**Page 8, line 220**. It seems peculiar that you have chosen to investigate theoretically only the CH2OO reaction and not the MVK-oxide reaction also. In MVK-oxide, the conjugation of the unsaturated side chain with the carbonyl oxide group has the potential to substantially

alter the surface. These calculations are likely significantly more complex than for the CH2OO case because of the need to consider syn and anti conformers, and cis/trans forms of each of these. However, given the interesting structural and conformeric dependence of Criegee intermediate reactivity, it is a regretful omission.

AUTHORS' REPLY:

Now we have the calculation result of MVKO+DMS reaction. Please see Reply to Referee 1 (for Line 220) for the calculation results on the direct reaction of MVKO + DMS, and the reply to referee 2 for catalysis reactions by DMS on unimolecular reactions of MVKO.

**Page 8, line 233**: A reference is needed to substantiate the statement regarding high barriers for DMS C-H insertion.

AUTHORS' REPLY:

We would add the paper of Decker et al. (*Phys. Chem. Chem. Phys.*, 2017,**19**, 8541-8551, doi:10.1039/C6CP08602K) in to the reference

**Page 8, line 251**: Do you anticipate stabilization of the (CH3)2SCH2OO adduct under tropospheric conditions?

The bonding is too weak to be stabilized under tropospheric conditions.

**Page 9, line 261:** You hypothesize that surface reactions converting DMS to SO2 in the chamber study of Newland could be the source of discrepancy between the present work and the work of Newland et al. I encourage the authors to respond to the comments of Rickard, Newland and Bloss, and Blitz regarding this matter.

AUTHORS' REPLY:

We would respond to the comments of Rickard et al. separately in the online discussion system of ACP.

**Figure 2**: Please include a note to address if the error bars are included or not included on this plot (as noted for Figure 4). Given that the rate coefficients for the self-reaction of CH2OO and the reaction of CH2OO + I (see Blitz review) are now well established, it would

be pertinent to deduce which of these is the major source of increased loss rates at higher laser fluence are under your experimental conditions.

AUTHORS' REPLY:

(i) We would add the following text in the caption:

*"For each data point, the error of the single exponential fitting is less than 1% (thus not shown)."*

(ii) Based on the absolute absorption cross section of $CH_2OO$ at 340 nm ($\sigma = 1.23\times10^{-17}$ cm$^2$) and the pressure-dependent yield of $CH_2OO$ from $CH_2I + O_2$ (0.46 at 300 Torr) (Ting et al., 2014a) the number densities of relevant species can be estimated to be the following (for Exp#1, Table S1).

$[CH_2OO]_0 = 6.7\times10^{11}$ cm$^{-3}$; $[I]_0 = 2.1\times10^{12}$ cm$^{-3}$; $[CH_2IOO]_0 = 7.7\times10^{11}$ cm$^{-3}$.

The first-order decay rate coefficient of $CH_2OO$ ($k_{eff}$) can be approximately estimated (Li et al., 2020) as:

$$k_{eff} = k_I[I]_0 + k_{self}[CH_2OO]_0$$

Using $k_{self} = 8\times10^{-11}$ cm$^3$ s$^{-1}$ and $k_I = 5.8\times10^{-11}$ cm$^3$ s$^{-1}$ at 300 Torr (Mir et al., 2020), the estimated $k_{eff}$ is 180 s$^{-1}$, consistent with the observed value of 232 s$^{-1}$ for $k_0$. Therefore, the main loss processes of $CH_2OO$ are reaction with iodine atoms (and other radicals) and its self-reaction.

In Figure S7, we can see a nice linear relationship between $k_0$ and the total produced radicals (proportional to the product of the laser fluence and the precursor concentration), further supporting the above mechanism. We would add the following sentences in the caption of Figure S7.

*"The main loss processes of $CH_2OO$ are reactions with radical byproducts like iodine atoms and its self-reaction. The observed values of $k_0$ (e.g., 232 s$^{-1}$ for Exp#1) are consistent with the values (e.g., 180 s$^{-1}$ at the condition of Exp#1) that are estimated using the reported kinetic data (yield and rate coefficients) (Mir et al., 2020; Ting et al., 2014)."*

**Supplementary information**

**Table S3:** Because both the reaction forming MVK-oxide from the precursor + O2 reaction

as well as the MVK-oxide + SO2 reaction features an adduct, the authors should label more caerefully the adduct referred to in the 'adduct yield' column of the table to avoid confusion.

AUTHORS' REPLY: We will add a footnote after the "adduct yield$^a$"

$^a$ *The yield of CH$_3$(C$_2$H$_3$)CIOO.*

**Figures S1, S2:** Provide details about error bars (c.f. comment about Figure 2).

AUTHORS' REPLY:

We would add the following text into the caption

*"For each data point, the error of the single exponential fitting is lees than 1% (thus not shown)."*

**Figure S4:** Please discuss the proposed origin of the "spike" at time zero.

AUTHORS' REPLY:

The photolysis of DMS produces radicals like CH$_3$ and CH$_3$S. A few vibronic bands of the A-X transition of the CH$_3$S radical (Liu et al., 2005) are within our probe window (335-345 nm). Thus it is possible that the "spike" near time zero is due to the absorption of the radical products of DMS photolysis, likely CH$_3$S or vibrationally excited CH$_3$S. We would add the following sentence in the figure caption.

*"The absorbance change under zero [DMS] comes from the interaction of the optics and the photolysis laser pulse, whereas the "spike" near time zero at high [DMS] may come from the absorption of the radical products of DMS photolysis, likely CH$_3$S (Liu et al., 2005) and/or vibrationally excited CH$_3$S."*

**S10:** These additional investigations are illuminating and interesting.

AUTHORS' REPLY: Thanks.

Additional comments regarding MVK-oxide conformers

I would like to add some discussion to the comments made by other reviewers regarding which conformers of MVK-oxide are produced from the photolytic scheme vs. ozonolysis. While the distribution of these conformers has not yet been deduced, the recent literature on

direct MVK-oxide kinetic and spectroscopic studies that indicated that both syn (Caravan et al., 2020) and anti (Vansco et al., 2020) confirmers are produced from the 1,3,-diiodobut-2-ene photolysis scheme used in the present work (Barber et al., 2018). Additionally, due to the rapid unimolecular decay of anti compared with syn (Barber et al., 2018;Vereecken et al., 2017), it is unlikely that reaction with DMS could compete with unimolecular decay under tropospheric conditions for the anti conformer.

AUTHORS' REPLY:

Same as the reply to Referee 2 (for Supplemental Information S20-S21), we have added some description to after line 80 to clarify the MVKO conformation.

"*For MVKO, there are 4 possible conformers. Following the nomenclature of Barber et al., syn/anti-MVKO (E/Z-MVKO) has a methyl/vinyl group at the same side of the terminal oxygen, while cis and trans refer to the orientation between the vinyl C=C and the carbonyl C=O bonds (Barber et al., 2018). It has been reported that syn- and anti-MVKO do not interconvert due to a high barrier between them but the barrier between cis and trans forms is low enough to permit fast interconversion at 298 K (Barber et al., 2018;Vereecken et al., 2017). Caravan et al., have shown that anti-MVKO is unobservable under thermal (298 K) conditions due to short lifetime and/or low yield, and thus, the UV-Vis absorption signal is from an equilibrium mixture of cis and trans forms of syn-MVKO (Caravan et al., 2020;Vereecken et al., 2017). For simplicity we will use MVKO to represent syn-MVKO (E-MVKO).*"

**References:**

Atkinson, R., Baulch, D. L., Cox, R. A., Crowley, J. N., Hampson, R. F., Hynes, R. G., Jenkin, M. E., Rossi, M. J., and Troe, J.: Evaluated kinetic and photochemical data for atmospheric chemistry: Volume I - gas phase reactions of $O_x$, $HO_x$, $NO_x$ and $SO_x$ species, Atmos. Chem. Phys., 4, 1461-1738, https://doi.org/10.5194/acp-4-1461-2004, 2004.

Bain, M., Hansen, C. S., and Ashfold, M. N. R.: Communication: Multi-mass velocity map imaging study of the ultraviolet photodissociation of dimethyl sulfide using single photon ionization and a PImMS2 sensor, J. Chem. Phys., 149, 081103, 10.1063/1.5048838, 2018.

Barber, V. P., Pandit, S., Green, A. M., Trongsiriwat, N., Walsh, P. J., Klippenstein, S. J., and Lester, M. I.: Four-Carbon Criegee Intermediate from Isoprene Ozonolysis: Methyl Vinyl Ketone Oxide Synthesis, Infrared Spectrum, and OH Production, J. Am. Chem. Soc., 140, 10866-10880, 10.1021/jacs.8b06010, 2018.

Caravan, R. L., Vansco, M. F., Au, K., Khan, M. A. H., Li, Y.-L., Winiberg, F. A. F., Zuraski, K., Lin, Y.-H., Chao, W., Trongsiriwat, N., Walsh, P. J., Osborn, D. L., Percival, C. J., Lin, J. J.-M., Shallcross, D. E., Sheps, L., Klippenstein, S. J., Taatjes, C. A., and Lester, M. I.: Direct kinetic measurements and theoretical predictions of an isoprene-derived Criegee intermediate, Proc. Natl. Acad. Sci., 117, 9733-9740, 10.1073/pnas.1916711117, 2020.

Cox, R. A., Ammann, M., Crowley, J. N., Herrmann, H., Jenkin, M. E., McNeill, V. F., Mellouki, A., Troe, J., and Wallington, T. J.: Evaluated kinetic and photochemical data for atmospheric chemistry: Volume VII - Criegee intermediates, Atmos. Chem. Phys. Discuss., 2020, 1-41, 10.5194/acp-2020-472, 2020.

Decker, Z. C. J., Au, K., Vereecken, L., and Sheps, L.: Direct experimental probing and theoretical analysis of the reaction between the simplest Criegee intermediate $CH_2OO$ and isoprene, Phys. Chem. Chem. Phys., 19, 8541-8551, 10.1039/C6CP08602K, 2017.

Hatakeyama, S., and Akimoto, H.: Reactions of criegee intermediates in the gas phase, Res. Chem. Intermed., 20, 503-524, 10.1163/156856794X00432, 1994.

Johnson, D., Lewin, A. G., and Marston, G.: The Effect of Criegee-Intermediate Scavengers on the OH Yield from the Reaction of Ozone with 2-methylbut-2-ene, J. Phys. Chem. A, 105, 2933-2935, 10.1021/jp003975e, 2001.

Johnson, D., and Marston, G.: The gas-phase ozonolysis of unsaturated volatile organic compounds in the troposphere, Chem. Soc. Rev., 37, 699-716, 10.1039/B704260B, 2008.

Khan, M. A. H., Cooke, M. C., Utembe, S. R., Archibald, A. T., Derwent, R. G., Xiao, P., Percival, C. J., Jenkin, M. E., Morris, W. C., and Shallcross, D. E.: Global modeling of the nitrate radical ($NO_3$) for present and pre-industrial scenarios, Atmospheric Research, 164-165, 347-357, https://doi.org/10.1016/j.atmosres.2015.06.006, 2015.

Kuwata, K. T., Guinn, E. J., Hermes, M. R., Fernandez, J. A., Mathison, J. M., and Huang, K.: A Computational Re-examination of the Criegee Intermediate–Sulfur Dioxide Reaction, J. Phys. Chem. A, 119, 10316-10335, 10.1021/acs.jpca.5b06565, 2015.

Li, M., Karu, E., Brenninkmeijer, C., Fischer, H., Lelieveld, J., and Williams, J.: Tropospheric OH and stratospheric OH and Cl concentrations determined from $CH_4$, $CH_3Cl$, and $SF_6$ measurements, npj Climate and Atmospheric Science, 1, 29, 10.1038/s41612-018-0041-9, 2018.

Li, Y.-L., Kuo, M.-T., and Lin, J. J.-M.: Unimolecular decomposition rates of a methyl-substituted Criegee intermediate *syn*-$CH_3CHOO$, RSC Advances, 10, 8518-8524, 10.1039/D0RA01406K, 2020.

Limão-Vieira, P., Eden, S., Kendall, P. A., Mason, N. J., and Hoffmann, S. V.: High resolution VUV photo-absorption cross-section for dimethylsulphide, $(CH_3)_2S$, Chemical Physics Letters, 366, 343-349, https://doi.org/10.1016/S0009-2614(02)01651-2, 2002.

Lin, Y.-H., Li, Y.-L., Chao, W., Takahashi, K., and Lin, J. J.-M.: The role of the iodine-atom adduct in the synthesis and kinetics of methyl vinyl ketone oxide—a resonance-stabilized Criegee intermediate, Phys. Chem. Chem. Phys., 22, 13603-13612, 10.1039/D0CP02085K, 2020.

Liu, C.-P., Reid, S. A., and Lee, Y.-P.: Two-color resonant four-wave mixing spectroscopy of highly predissociated levels in the $\tilde{A}A_{12}$ state of $CH_3S$, J. Chem. Phys., 122, 124313, 10.1063/1.1867333, 2005.

Mir, Z. S., Lewis, T. R., Onel, L., Blitz, M. A., Seakins, P. W., and Stone, D.: $CH_2OO$ Criegee intermediate UV absorption cross-sections and kinetics of $CH_2OO + CH_2OO$ and $CH_2OO + I$ as a function of pressure, Phys. Chem. Chem. Phys., 22, 9448-9459, 10.1039/D0CP00988A, 2020.

Newland, M. J., Rickard, A. R., Vereecken, L., Muñoz, A., Ródenas, M., and Bloss, W. J.: Atmospheric isoprene ozonolysis: impacts of stabilised Criegee intermediate reactions with $SO_2$, $H_2O$ and dimethyl sulfide, Atmos. Chem. Phys., 15, 9521-9536, 10.5194/acp-15-9521-2015, 2015.

Smith, M. C., Chao, W., Takahashi, K., Boering, K. A., and Lin, J. J. M.: Unimolecular Decomposition Rate of the Criegee Intermediate $(CH_3)_2COO$ Measured Directly with UV Absorption Spectroscopy, J. Phys. Chem. A, 120, 4789-4798, 10.1021/acs.jpca.5b12124, 2016.

Taatjes, C. A., Meloni, G., Selby, T. M., Trevitt, A. J., Osborn, D. L., Percival, C. J., and Shallcross, D. E.: Direct Observation of the Gas-Phase Criegee Intermediate $(CH_2OO)$, J. Am. Chem. Soc., 130, 11883-11885, 10.1021/ja804165q, 2008.

Ting, W. L., Chang, C. H., Lee, Y. F., Matsui, H., Lee, Y. P., and Lin, J. J. M.: Detailed mechanism of the $CH_2I + O_2$ reaction: Yield and self-reaction of the simplest Criegee intermediate $CH_2OO$, J. Chem. Phys., 141, 104308, 10.1063/1.4894405, 2014a.

Ting, W. L., Chen, Y. H., Chao, W., Smith, M. C., and Lin, J. J. M.: The UV absorption spectrum of the simplest Criegee intermediate $CH_2OO$, Phys. Chem. Chem. Phys., 16, 10438-10443, 10.1039/c4cp00877d, 2014b.

Vansco, M. F., Caravan, R. L., Zuraski, K., Winiberg, F. A. F., Au, K., Trongsiriwat, N., Walsh, P. J., Osborn, D. L., Percival, C. J., Khan, M. A. H., Shallcross, D. E., Taatjes, C. A., and Lester, M. I.: Experimental Evidence of Dioxole Unimolecular Decay Pathway for Isoprene-Derived Criegee Intermediates, J. Phys. Chem. A, 124, 3542-3554, 10.1021/acs.jpca.0c02138, 2020.

Vereecken, L., Novelli, A., and Taraborrelli, D.: Unimolecular decay strongly limits the atmospheric impact of Criegee intermediates, Phys. Chem. Chem. Phys., 19, 31599-31612, 2017.

Welz, O., Savee, J. D., Osborn, D. L., Vasu, S. S., Percival, C. J., Shallcross, D. E., and Taatjes, C. A.: Direct Kinetic Measurements of Criegee Intermediate $(CH_2OO)$ Formed by Reaction of $CH_2I$ with $O_2$, Science, 335, 204-207, 10.1126/science.1213229, 2012.

---

## Author Comment (AC2) · 27 Aug 2020

We thank Rickard et al. for helpful comments. Our reply is given below.

**AUTHORS' REPLY to SC1:**

We fully agree the following comments by Rickard et al. (DOI:10.5194/acp-2020-484-SC1):

"... the isoprene-ozone system is significantly more complex than these previously studied ozonolysis systems, with a range of different SCI species formed, with different yields and exhibiting different bimolecular and unimolecular kinetics."

Rickard et al. are right about that there are various stabilized CIs (CH2OO, MVKO, and MACRO) generated in the isoprene-ozone system. In addition, there are multiple conformers (*syn/anti* and *cis/trans*) for MVKO or MACRO. "Therefore 9 different types and configurations of SCI can be formed in the isoprene ozonolysis system" as Rickard et al. have mentioned (DOI:10.5194/acp-2020-484-SC1). For this issue, we would add the following sentences to clarify (after Line 80).

"For MVKO, there are 4 possible conformers. Following the nomenclature of Barber et al., *syn/anti*-MVKO has a methyl/vinyl group at the same side of the terminal oxygen, while *cis* and *trans* refer to the orientation between the vinyl C=C and the carbonyl C=O bonds (Barber et al., 2018). It has been reported that *syn-* and *anti-*MVKO do not interconvert due to a high barrier between them but the barrier between *cis* and *trans* forms is low enough to permit fast interconversion at 298 K (Barber et al., 2018; Vereecken et al., 2017). Caravan et al., have shown that *anti-*MVKO is unobservable under thermal (298 K) conditions due to short lifetime and/or low yield, and thus, the UV-Vis absorption signal is from an equilibrium mixture of *cis* and *trans* forms of *syn-*MVKO (Caravan et al., 2020). For simplicity we will use MVKO to represent *syn-*MVKO in this work."

The conformer populations in the ozonolysis system may differ from those in the photolytic generation of MVKO using the diiodo precursor. However, the thermal lifetime of *anti*-MVKO (< 0.5 ms) (Vereecken et al., 2017; Barber et al., 2018) would be too short to support any significant concentration of *anti*-MVKO for participating in bimolecular reactions. For the remaining longer-lived *cis* and *trans* conformers of *syn*-MVKO, fast interconversion between the *cis* and *trans* forms would lead to an equilibrium mixture in both cases of ozonolysis and photolytic generation of CIs. In this regard, the conformer

populations of stabilized *syn*-MVKO would be similar in both cases. Thus, our direct kinetic results can be used to constrain the relevant reaction pathways for the ozonolysis systems.

More importantly, as already stated in the paper of Newland et al. (Newland et al., 2015),

"However, it is important to note that no constraints regarding the products of the proposed DMS + SCI reaction were obtained; OH reaction with DMS is complex, proceeding through both abstraction and addition/complex formation channels, the latter rendered partially irreversible under atmospheric conditions through subsequent reaction with O2 (Sander et al., 2011). The observed behaviour (Fig. 5) is not consistent with reversible complex formation dominating the SCI-DMS system under the conditions used; however it is possible that decomposition of such a complex to reform DMS, or its further reaction (e.g. with SO2, analogous to the secondary ozonide mechanism proposed by Hatakeyama et al., 1986), would be consistent with the observed data, and also imply that the reaction may not lead to net DMS removal. Time-resolved laboratory measurements and product studies are needed to provide a test of this mechanistic possibility."

Indeed, there are multiple possibilities in the complicated isoprene-ozone system. Again our current results for individual CIs may provide useful constraints for such systems.

We agree that due to the large volume to surface ratio in the experiments of Newland et al., surface reaction may not be very important. We would revise the relevant text (after Line 212) to:

"For the determination of the relative rate of the CI + DMS reaction, Newland et al. monitored the consumption of SO2 over a measurement period of up to 60 min until approximately 25% of isoprene was consumed (Newland et al., 2015). Additional uncharacterized reaction pathways (e.g., reactions with the products and/or catalytic reactions) would lead to a bias in the inferred rate coefficients. A part of this high complexity of the isoprene-ozone-DMS-SO2 system has been discussed by Newland et al. in the section of Experimental Uncertainties (Newland et al., 2015). Our direct measurements and kinetics are very straightforward; the obtained results for individual CIs may provide useful constraints for related ozonolysis systems."

2

Finally, an interesting aspect raised by a referee (Referee 1) is:

"As this other study generated the Criegees via ozonolysis (O3/isoprene), it does ask the question how we best understand ozonolysis in the atmosphere. Is stabilized Criegee chemistry the most important component of ozonolysis?"

This is possible. But we only generated individual CIs using a well-established photolytic method and cannot comment or answer this question.

**References:**

Barber, V. P., Pandit, S., Green, A. M., Trongsiriwat, N., Walsh, P. J., Klippenstein, S. J., and Lester, M. I.: Four-Carbon Criegee Intermediate from Isoprene Ozonolysis: Methyl Vinyl Ketone Oxide Synthesis, Infrared Spectrum, and OH Production, J. Am. Chem. Soc., 140, 10866-10880, 10.1021/jacs.8b06010, 2018.

Caravan, R. L., Vansco, M. F., Au, K., Khan, M. A. H., Li, Y.-L., Winiberg, F. A. F., Zuraski,
K., Lin, Y.-H., Chao, W., Trongsiriwat, N., Walsh, P. J., Osborn, D. L., Percival, C. J., Lin, J.
J.-M., Shallcross, D. E., Sheps, L., Klippenstein, S. J., Taatjes, C. A., and Lester, M. I.: Direct kinetic measurements and theoretical predictions of an isoprene-derived Criegee intermediate,
Proc. Natl. Acad. Sci., 117, 9733-9740, 10.1073/pnas.1916711117, 2020.

Newland, M. J., Rickard, A. R., Vereecken, L., Muñoz, A., Ródenas, M., and Bloss, W. J.: Atmospheric isoprene ozonolysis: impacts of stabilised Criegee intermediate reactions with SO2, H2O and dimethyl sulfide, Atmos. Chem. Phys., 15, 9521-9536, https://doi.org/10.5194/acp-15-9521-2015, 2015.

Vereecken, L., Novelli, A., and Taraborrelli, D.: Unimolecular decay strongly limits the atmospheric impact of Criegee intermediates, Phys. Chem. Chem. Phys., 19, 31599-31612, 2017.

---

## Author Response (AR1)

We thank the referees for their careful reading of the manuscript and helpful comments, which are repeated below (in black font). Our replies are given in blue font directly after each comment.

**Referee 1:**

**General comments**

Time-resolved experiments have been carried out to generate two Criegee: $CH_2OO$, which has been study many times; and MVKOO, which has only very recently been studied. In the presence of dimethyl sulphide, no additional Criegee removal was evident. Hence, only an upper limit is assigned for the rate coefficients. A theoretical potential energy surface has been calculated for $CH2OO + (CH3)2S$ that has a significant barrier to products (DMSO + $CH_2O$) and its rate coefficient is lower than the experimental upper limit. These results are clear-cut and only a few specific comments are raised.

If this were the only study on the titled reaction, the lack of reactivity would probably mean this paper would not be considered for publication in ACP. The reason this result is significant is that a previous study (Newland 2015) suggested the stabilized Criegee formed from $O_3$/isoprene (mainly $CH_2OO$/MKVO) react rapidly with dimethyl sulphide, with a rate coefficient close to the gas-kinetic frequency. As this other study generated the Criegees via ozonolysis ($O_3$/isoprene), it does ask the question how we best understand ozonolysis in the atmosphere. Is stabilized Criegee chemistry the most important component of ozonolysis? More detail would help this paper. The comment from Andrew Rickard expands on this.

**Specific comments**

**Line 39** "*Surprisingly, the obtained rate coefficients are up to $10^4$ times larger than previous results deduced from ozonolysis experiments, indicating that the ozonolysis experiments could be quite complicated such that reliable kinetic results may be hard to retrieve.*"

This needs a reference. This is interesting in that relative rate experiments appear to be out by orders of magnitude. Is there explanation of these studies with today's knowledge? Is it wrong rate coefficients or is it more to do with the experiment itself?

AUTHORS' REPLY:

Welz et al. (2012) compared their rate coefficients with previous values applied in contemporary tropospheric models (Johnson and Marston, 2008; Johnson et al., 2001; Hatakeyama and Akimoto, 1994). For ozonolysis experiments, typically only the ratios of reaction rate coefficients, e.g. $k_{DMS}/k_{SO2}$ (Newland et al., 2015), are obtained. The researchers have to compare with (at least) one absolute rate coefficient to get the rest rate coefficients. Unfortunately, the selected absolute rate coefficient (at that time) has large uncertainty, which propagates to other reported values. In addition, the reaction mechanism may be rather complicated and even the ratios of the rate coefficients must be treated with care. The above three references will be included in the main text.

**Line 103** "*To compensate for this effect, which was caused by the optics and the photolysis laser pulse, we recorded background traces without adding the precursor before and after each set of experiments. The reported data are after background subtraction.*" Can you state the typically size of this signal, i.e. what is I/I0 in the absence of added chemicals. Is it related to a heating effect?

AUTHORS' REPLY:

Typical background traces as well as raw signal traces (without background subtraction) obtained at 248 nm and 308 nm will be shown in Figures S5 and S6, respectively. These backgrounds are originated from the different longpass filters used for coupling the laser beam and probe beam into the reactor. Yes, it is likely that the backgrounds are from a heating effect of the longpass filters.

[Figure]

Fig. S5. Background traces under normal DMS concentrations, represented in colour lines, and the raw signal traces (without background subtraction), represented in grey lines, obtained with 248 nm photolysis laser ($I_{248nm}$ = 2.43 mJ cm$^{-2}$). See Exp#22 of Table S3 for the experimental conditions.

[Figure]

Fig. S6. Background traces under normal DMS concentrations, represented in colour lines, and the raw signal traces (without background subtraction), represented in grey lines, obtained with 308 nm photolysis laser ($I_{308nm}$ = 2.35 mJ cm$^{-2}$). See Exp#2 of Table S1 for the experimental condition. Note that the optics (longpass filters) are different from those at 248 nm.

**Line 134** "*e.g., bimolecular reactions with radical byproducts like I atoms, wall loss, etc.*" Probably self-reaction is most important. Any evidence for a second-order component?

This paper has probably done most to unravel the removal the kinetics in absence of added

reagent. CH2OO Criegee intermediate UV absorption cross-sections and kinetics of CH2OO + CH2OO and CH2OO + I as a function of pressure By:Mir, ZS (Mir, Zara S.)[ 1 ] ; Lewis, TR (Lewis, Thomas R.)[ 1 ] ; Onel, L (Onel, Lavinia)[ 1 ] ; Blitz, MA (Blitz, Mark A.)[ 1,2 ] ; Seakins, PW (Seakins, Paul W.)[ 1 ] ; Stone, D (Stone, Daniel)[ 1 ]

AUTHORS' REPLY:

In the previous works of Smith et al. (Smith et al., 2016) and Li et al. (Li et al., 2020), the contributions of the pseudo-first-order reactions and second-order reactions are both considered and the kinetic model can be represented in the following equation:

$$\frac{-d[\text{CI}]}{dt} = k_1[\text{CI}] + k_2[\text{CI}]^2$$

The above equation can be simplified when extrapolating the rate coefficients to zero concentration of $[\text{CI}]_0$:

$$\frac{-d[\text{CI}]}{dt} \cong (k_1 + \frac{1}{2}k_2[\text{CI}]_0)[\text{CI}] = k_{\text{obs}}[\text{CI}]$$

The difference between the complete and simplified equations only shows up at high $[\text{CI}]_0$. Most important of all, the self-reaction of CIs would not affect the determination of $k_{\text{DMS}}$, since $[\text{CI}]_0$ was kept constant in every experimental set.

Based on the absolute absorption cross section of $CH_2OO$ at 340 nm ($\sigma = 1.23 \times 10^{-17}$ cm$^2$) and the pressure-dependent yield of $CH_2OO$ from $CH_2I + O_2$ (0.46 at 300 Torr) (Ting et al., 2014a) the number densities of relevant species can be estimated to be the following (for Exp#1, Table S1).
$[CH_2OO]_0 = 6.7 \times 10^{11}$ cm$^{-3}$; $[I]_0 = 2.1 \times 10^{12}$ cm$^{-3}$; $[CH_2IOO]_0 = 7.7 \times 10^{11}$ cm$^{-3}$.
The first-order decay rate coefficient of $CH_2OO$ ($k_{\text{eff}}$) can be approximately estimated (Li et al., 2020) as:

$$k_{\text{eff}} = k_I[I]_0 + k_{\text{self}}[CH_2OO]_0$$

Using $k_{\text{self}} = 8 \times 10^{-11}$ cm$^3$ s$^{-1}$ and $k_I = 5.8 \times 10^{-11}$ cm$^3$ s$^{-1}$ at 300 Torr (Mir et al., 2020), the estimated $k_{\text{eff}}$ is 180 s$^{-1}$, consistent with the observed value of 232 s$^{-1}$ for $k_0$. Therefore, the main loss processes of $CH_2OO$ are reaction with iodine atoms (and other radicals) and its self-reaction.

In Figure S7, we can see a nice linear relationship between $k_0$ and the total produced radicals (proportional to the product of the laser fluence and the precursor concentration), further supporting the above mechanism. We would add the following sentences in the caption of Figure S7.

*"The main loss processes of $CH_2OO$ are reactions with radical byproducts like iodine atoms and its self-reaction. The observed values of $k_0$ (e.g., 232 $s^{-1}$ for Exp#1) are consistent with the values (e.g., 180 $s^{-1}$ at the condition of Exp#1) that are estimated using the reported kinetic data (yield and rate coefficients) (Mir et al., 2020; Ting et al., 2014)."*

We will also modify the relevant sentences in the main text to:

*"The subsequent decay in absorption is due to the consumption of $CH_2OO$ either through reaction with DMS or through other processes, e.g., bimolecular reactions with radical byproducts like I atoms, wall loss, etc. In addition, self-reaction of $CH_2OO$ has been found to be rather fast ($k_{self} = 8 \times 10^{-11} cm^3 s^{-1}$)(Mir et al., 2020). However, the effect of the self-reaction (Smith et al., 2016; Li et al., 2020) would not affect the determination of $k_{DMS}$ under our experimental conditions."*

**Line 155** "*and show the results in Table S4.*" , From Table S4, the results given in Figure 2 are fairly obvious. I would expected a similar result even if 248 nm photolysis was used. Significant photolysis of DMS could potentially lead to enhanced reactivity, and an energy dependence would be good practice. However, in the present case, there is no evidence of enhanced Criegee removal so there is not too much to worry about.

AUTHORS' REPLY:

In Table S4, we have shown that $[DMS]_{diss}$ is about ten times less than $[CH_2I_2]_{diss}$ under typical experimental conditions when 248 nm photolysis is applied. However, we have observed a strong absorption in the background traces when 248 nm photolysis and high [DMS] are applied (Figure S4). The extra absorption from the dissociated DMS would be problematic when performing the background subtraction. Therefore we constrained the laser fluence and [DMS] to preclude the influence of [DMS] photolysis.

**Line 171** "*See SI (Sect. S3, page S5) for details.*" From the SI (Table S3), the instant yield of MVKOO decreases with total pressure, which is consistent with population into $CH_3(C_2H_3)CIOO$, i.e. the SV is linear. However, $k_r$ appears to be faster at low pressures. $k_r$ is the rate coefficient for the peroxy radical to react to MVKOO + I. It is not possible for a rate coefficient to increase at lower total pressure. There are too few pressures to say anything for definite, but it does highlight that the $k_r$ errors are not realistic.

I wonder if there is another explanation for the results in Table S3. If you had an additional species, produced from the photolysis of the di-iodo compound, X, that can react with the di-iodo compound to make the iodo radical.

di-iodo + hv $\rightarrow$ X

X + di-iodo $\rightarrow$ iodo radical

The pressure dependence could be linked to the fact the MVKOO species has a double bond.

If $k_r$ is the unimolecular reaction $CH_3(C_2H_3)CIOO \rightarrow MVKOO + I$, then changing the temperature should be the easiest way to identify it.

**Line 172** "*This difference is consistent with the fact that MVKO is resonance-stabilized due to the extended conjugation of its vinyl group (Barber et al., 2018) and thus the adduct $CH_3(C_2H_3)CIOO$ is relatively less stable due to disruption of the conjugation.*"

It will be the properties of $CH_3(C_2H_3)CIOO$ that will most strongly influence its formation and unimolecular dissociation, $k_r$.

AUTHORS' REPLY (To lines 171-172):

The reviewer is right about the role of $CH_3(C_2H_3)CIOO$ and that changing the temperature should be the easiest way to identify the related process. In fact, we have discussed the issues of the adduct, including the temperature and pressure effects, in our recent paper (Lin et al., 2020). Since MVKO is a resonance-stabilized molecule, adduct would be relatively less stable, compared with CIs without resonance structure, such as $CH_2OO$ or $CH_3CHOO$. Therefore, the unimolecular decomposition of the adduct is observed in our experimental time scale. The reason why $k_r$ appears to be larger at lower pressure is that the fitted $k_r$ should include the unimolecular decomposition of the adduct and the reaction of the adduct with other radicals (X) such as iodine atoms.

$$k_r = k_{uni}(adduct) + k_x[X]$$

The concentration of the radicals would increase as the precursor concentration increases, leading to a higher $k_r$. This relation can be observed explicitly through plotting $k_r$ against $I_{248 nm} \times Abs$(238 nm) (photolysis laser fluence times precursor absorbance). As for the temperature effect, we have also observed a positive temperature dependence of $k_r$ ($E_a$ = 12.7±0.3 kcal mol$^{-1}$), consistent with the calculation result for the bond dissociation energy of the adduct (14 kcal mol$^{-1}$) (Lin et al., 2020).

[Figure]

Fig. S?. Plot of $k_r$ against the product of the laser fluence ($I_{248nm}$) and the absorbance of 1,3-diiodo-2-butene at 238 nm in the photolysis cell ($Abs$(238nm)) for the experiments of MVKO+DMS reaction (Exp#15-29, Tables S3). The x-axis essentially represents the total amounts of radical species generated through the photolysis of the precursor (R1) and the subsequent reactions (R2). Higher radical concentration results in faster decay of the adduct, thus higher $k_r$.

Please note that the error bars in Tables S1-S3 do NOT include any systematic errors. For $k_r$, it is correlated with other fitting parameters like (1−$\alpha$). Since MVKO does not react with DMS (essentially all the traces are almost the same at various [DMS]), it is hard to 'disentangle' the correlation among fitting parameters. In the paper by Lin et al., we used SO$_2$ to scavenge MVKO and to obtain more robust results (Lin et al., 2020).

We will add a notation regarding the error bar of $k_r$ after Table S3:

"*averaged value ± 1 sigma error of the mean (statistical only, not including systematic*

*errors). The actual error bar would be larger since $k_r$ is highly correlated with other fitting parameters like (1−α). Lin et al. has used SO₂ scavenger to obtain more robust results for $k_r$ (Lin et al., 2020)."*

**Line 194** "*Here we choose the boundary of three standard deviations as the upper limits for $k_{DMS}+CI$, $k_{DMS}+CH_2OO \leq 4.2\times10^{15}$ cm³s⁻¹ and $k_{DMS}+MVKO \leq 1.6\times10^{14}$ cm³s⁻¹*" As you have done calculations, it would be better to state that the expts provide only an upper limit, and it is most likely that the *k* are smaller and closer to the theoretical values.

AUTHORS' REPLY

Indeed, the actual value of $k_{DMS+CI}$ would be smaller than the upper limits we reported, and the actual value of $k_{DMS+CH2OO}$ may be closer to the theoretical value ($k_{DMS+CH2OO} = 5.5\times10^{-19}$ cm³ s⁻¹). However, the calculation is not at the best level (while it is still good enough for the discussion in this paper) and there are uncertainties in the calculated values. Thus we decided not to say that the rate coefficients would be closer to the theoretical values.

**Line 203** "*[CI]ss is expected to be low, at least a couple of orders of magnitude lower than the steady-state [OH]ss.*" On this basis, reactions need to be two orders of magnitude faster than OH to compete. SO2, H2O vapour and acids fit the bill but not many other reagents.

AUTHORS' REPLY

We totally agree with your point. Thus we think the reaction of CI+DMS would not be a major path for the oxidization of DMS since the rate coefficient of CI+DMS is quite small.

**Line 216** "*While our direct measurements and kinetics are very straightforward, the ozonolysis experiments of Newland et al. might have been more complex than the authors (Newland et al., 2015) had assumed. For example, one may consider the possibility of converting DMS to SO2 via surface or gas-phase reactions (Chen et al., 2018) under the complicated conditions of isoprene ozonolysis.*"

Is this a reasonable conclusion? In the introduction, you mentioned that prior to direct timeresolved experiments, Criegee + $SO_2$ rate coefficients were thought to be slow. Is this another example of "surface" reactions? Is there more going in these ozonolysis experiments that bring about chemical change that if not captured by these direct measurements. Or are these relative rate reactions simply flawed?

Are there any suggested DMS $\rightarrow$ $SO_2$ schemes via the gas-phase?

AUTHORS' REPLY

The reviewer raised a few important and interesting questions, which are awaiting more investigations. As mentioned before, researchers have to postulate the reaction mechanism of the ozonolysis reaction to deduce the rate coefficients. We believe there are more to be studied for the ozonolysis of isoprene. For clarification, we would modify the related text to the following.

*"For the determination of the relative rate of the CI + DMS reaction, Newland et al. monitored the consumption of $SO_2$ over a measurement period of up to 60 min until approximately 25% of isoprene was consumed (Newland et al., 2015). Additional uncharacterized reaction pathways (e.g., reactions with the products) would lead to a bias in the inferred rate coefficients. A part of this high complexity of the isoprene-ozone-DMS-$SO_2$ system has been discussed by Newland et al. in the section of Experimental Uncertainties (Newland et al., 2015). Our direct measurements and kinetics are very straightforward; the obtained results for individual CIs may provide useful constraints for related ozonolysis systems."*

**Line 220** Any reason why MKVOO + $SO_2$ not calculated?

AUTHORS' REPLY

We guess the reviewer meant MVKO+DMS. Now we have the calculation result of MVKO+DMS reaction. Similar to the reaction with $H_2O$ (Vereecken et al., 2017 ), the direct reaction of *E*- and *Z*-MVKO with DMS is expected to be slower than for $CH_2OO$, as the organic groups and the conjugation of the carbonyl oxide moiety with the double bond stabilizes the CI. Indeed, for MVKO (all conformers), no adduct seems to exist at the M06-2X/cc-pVDZ level of theory: the needed C−S bond in the adduct appears to be too weak to

compensate for the loss of the conjugation in the carbonyl oxide, and the system reverts to the MVKO + DMS complex instead, without a formal C−S bond. As a result, the barrier for the migration of a DMS methyl H-atom to the carbonyl oxide oxygen to form a methylidene adduct is ~10 kcal/mol higher than for the analogous TS in the $CH_2OO+DMS$ system which does feature a weakly bonded intermediate adduct. The direct oxygen transfer from *E*- or *Z*-MVKO to DMS, forming MVK + DMSO, was found to have a similarly high energy barrier as in the $CH_2OO+DMS$ system. No viable reaction channels were found involving the double bond in MVKO. The lack of accessible transition states then prohibits rapid direct reaction between DMS and MVKO.

We also have additional calculation on the cyclisation of MVKO in the presence of DMS. Again, no accessible pathways were found.

**TYPOS / UNDERSTANDING**

Is it MKVO or MVKOO? I think the later. This occurs several times

Also, MACRO or MACROO?

AUTHORS' REPLY

MVKO is short for methyl-vinyl-ketone-oxide, and is the correct notation (i.e. MVK + 1 oxide O-atom). Likewise, MACRO is an acronym for methacroleine-oxide. We have standardized on these notations, consistent with our previous paper (Lin et al., 2020).

**Line 74** "ozonlolysis" Typo

AUTHORS' REPLY

(will be fixed). Thanks for your reminder.

**Line 91** "However, DMS absorbs weakly at 248 nm. We therefore performed additional experiments by photolyzing CH2I2 at 248 nm to assess the impact of DMS photolysis at 248 nm on the decay of the CIs."

Do you mean 308 nm?

AUTHORS' REPLY

We want to emphasize that DMS absorbs weakly at 248 nm ($\sigma = 1.28\times10^{-20}$ cm$^2$) but barely absorbs at 308 nm ($\sigma < 1\times10^{-22}$ cm$^2$) (Limão-Vieira et al., 2002). At low [DMS], the weak absorption of DMS at 248 nm may not cause a problem, but in this work, [DMS] is quite high and thus the photolysis of DMS at 248 nm should be taken into consideration.

**Line 63** "*Newland et al. noted, however, that the presented rate coefficients do not correspond to the rates of single elementary reactions but rather describe the general reactivity of CIs towards DMS or H2O*" Can you re-phrase this as I'm not sure the point you making, be more explicit.

AUTHORS' REPLY:

Thank you for pointing out. The sentences will be rephrased to

"*Newland et al., who used ozonolysis of isoprene to generate a mixture of CIs (CH$_2$OO, MVKO, and MACRO), reported a combined reactivity of these CIs toward DMS and H$_2$O under conditions similar to the atmospheric boundary layer. Their reported rate coefficients might not correspond to those of single elementary reactions.*"

**Line 36** Beames et al., 2013 This is a depletion experiment.

AUTHORS' REPLY:

Thank you for pointing out. The sentence will be rephrased to

"*... UV-visible absorption/depletion spectroscopy ...*"

**Anonymous referee 2:**

One reason for the difference is the current results and the results reported in Newland 2015 may be the impact of DMS on the MVKO + $SO_2$ reaction. It is not necessary to perform calculations on this reaction, but some mechanistic discussion would be pertinent.

AUTHORS' REPLY:

The reactions of carbonyl oxides (CI) with $SO_2$ proceed by a barrierless cycloaddition (Kuwata et al., J. Phys. Chem. A, 119, 10316, 2015) with a very fast capture rate coefficient for complex formation near the collision limit, and a partial redissociation to the free reactants leading to a rate coefficient somewhat below the collision limit. The DMS-complex of a CI reacting with $SO_2$ can be expected to have a lower rate coefficient than the direct CI+$SO_2$ reaction, as the DMS shields part of the approach vectors of the $SO_2$ reactant, and the long-range attractive force is diminished due to a somewhat lower dipole moment of the complex compared to the free CI. However, the reduction of the rate coefficient is not expected to be all that large, and more importantly the CI+DMS complex is not overly strong such that only a small fraction of the CI will be present as a CI+DMS complex. This makes it hard to understand how DMS could affect any CI+$SO_2$ capture reaction ($CH_2OO$, MVKO, or $CH_3CHOO$) to the extent observed in Newland et al. It is for this reason that we have done exploratory calculations on the redissociation of the CI+$SO_2$ cyclo-adduct, but have found no indication that this would have the required impact on the effective CI+DMS rate of product formation.

**Line 224**: What is the evidence for the $CH_2OO$-DMS adduct having "very strong zwitterionic character?"

AUTHORS' REPLY:

At the level of theory used here, the wavefunction for the adduct converges to a closed-shell structure with no biradical character, where the O-atoms have a strongly negative partial charge (up to -0.46 in the Mulliken population analysis), and where the S-atom is positively charged S-atom (+0.28 in the Mulliken population analysis, compared to the Mulliken partial charge of -0.06 in DMS). This suggests that the $CH_2OO$-DMS adduct, similar to the parent

carbonyl oxide, has a zwitterionic character with very strong charge separation between the S and O atoms, rather than a biradical wavefunction.

**Supplemental Information S20-S21**: The authors should present some calculations on the MVKO. In particular, it would be worthwhile to consider how DMS might affect the cyclization of the anti conformer of MVKO to the dioxole (see J. Am. Chem. Soc. 2018, 140, 10866). Here, I reiterate the comment of Rickard that it would be useful for the authors to estimate the relative amounts of the syn and anti conformers of MVKO.

AUTHORS' REPLY:

The dominant unimolecular reaction of $E$-MVKO is a 1,4-H-shift (VHP-channel), analogous to $Z$-CH$_3$CHOO, for which we already showed that any catalytic effect is insufficient to allow for fast reactions. We now also calculated the impact of a DMS spectator complexing agent on the cyclization in $Z$-MVKO at the M06-2X/cc-pVDZ level of theory, finding similar results as for the methylated CH$_3$CHOO, i.e. the barrier height without (12.1 kcal/mol) and with complexing DMS (14.2 kcal/mol from the ground state of the complex) are essentially identical. The complex stability for $Z$-MVKO + DMS ($-9.9$ kcal/mol) is also similar to that for CH$_2$OO, $Z$-CH$_3$CHOO, and $E$-CH$_3$CHOO. Any catalyzing effect by DMS would then be due to chemical activation by the energy released in the complexation. The net energy barrier for the DMS catalysed $Z$-MVKO unimolecular reaction is ~ +4 kcal/mol, then still implies a slow bimolecular reaction, in agreement with the experimental observations.

Also see Reply to Referee 1 (for Line 220) for the calculation results on the direct reaction of MVKO + DMS.

Regarding the relative amounts of the *syn* and *anti* conformers of MVKO, we would add the following sentences to clarify the MVKO conformation. (after line 80)

*"For MVKO, there are 4 possible conformers. Following the nomenclature of Barber et al., syn/anti-MVKO (E/Z-MVKO) has a methyl/vinyl group at the same side of the terminal oxygen, while cis and trans refer to the orientation between the vinyl C=C and the carbonyl C=O bonds (Barber et al., 2018). It has been reported that syn- and anti-MVKO do not interconvert due to a high barrier between them but the barrier between cis and trans forms is low enough to permit fast interconversion at 298 K (Barber et al., 2018;Vereecken et al., 2017). Caravan et al., have shown that anti-MVKO is unobservable under thermal (298 K) conditions due to short lifetime and/or low yield, and thus, the UV-Vis absorption signal is from an equilibrium mixture of cis and trans forms of syn-MVKO (Caravan et al., 2020;Vereecken et al., 2017). For simplicity we will use MVKO to represent syn-MVKO (E-MVKO)."*

**Lines 232-233**: *"We did not examine more exotic CI reaction such as insertion in the DMS C–H bonds, as these are known to have comparatively high barriers."* This statement should have a reference.

AUTHORS' REPLY:

We would add the paper of (Decker et al. ***Phys. Chem. Chem. Phys.***, 2017,**19**, 8541-8551, doi:10.1039/C6CP08602K) into the reference

**Supplemental Information S20-S21**: The authors should tabulate the relative energies predicted by the M06-2X/cc-pVDZ calculations.

AUTHORS' REPLY:

A table is now included in the supporting information.

Table S_: ZPE-corrected DMS complex energies, E(complex), and barrier heights $E_b$ without and with a DMS complexing agent, at the M06-2X/cc-pVDZ level of theory. Energies are in kcal mol$^{-1}$ and relative to the free reactants.

| CI reaction | $E_b$ | E(complex) | $E_b$(complex) |
|---|---|---|---|
| $CH_2OO \rightarrow cyc\text{-}CH_2OO\text{-}$ | 22.0 | -9.6 | 14.5 |
| $Z\text{-}CH_3CHOO \rightarrow CH_2CHOOH$ | 12.7 | -8.6 | 7.2 |
| $Z\text{-}CH_3CHOO \rightarrow cyc\text{-}CH(CH_3)OO\text{-}$ | 25.8 | -8.6 | 18.2 |
| $E\text{-}CH_3CHOO \rightarrow cyc\text{-}CH(CH_3)OO\text{-}$ | 18.4 | -10.9 | 9.5 |
| $Z\text{-}(CH{=}CH_2)C(CH_3)OO \rightarrow cyc\text{-}CH\text{-}CH_2C(CH_3)OO\text{-}$ | 12.1 | -9.9 | 4.4 |
| $Z\text{-}(CH{=}CH_2)C(CH_3)OO + DMS \rightarrow MVK + DMSO$ | 8.7 | | |
| $E\text{-}(CH_3)C(CH{=}CH_2)OO + DMS \rightarrow MVK + DMSO$ | 8.0 | | |
| $(CH_3)C(CH{=}CH_2)OO + DMS \rightarrow$ $S(CH_3)({=}CH_2)C(CH_3)(CH{=}CH_2)OOH$ | 11.2 | | |

**Anonymous Referee #3**

Kuo et al. report direct experimental and theoretical investigations of the reactions of two isoprene-derived Criegee intermediates with dimethyl sulfide (DMS). Using the diiodoalkane/diiodoalkene photolysis method to selectively generate each Criegee intermediate in turn, the authors probe the kinetics by UV absorption and deduce upper limit rate coefficients that are orders of magnitude slower than those obtained in the ozonolysis work of Newland et al. using the relative rate technique. The slow rate coefficient measured in the present work for $CH_2OO$ + DMS is substantiated by stationary point calculations coupled with CTST that yield a rate coefficient of 5.5E-19 cm-3 s-1 at 298 K.

The paper is reasonably thorough and raises interesting discussion about ozonolysis vs. direct Criegee intermediate experimental kinetic studies, that have been significantly expanded by the other reviewers. The paper would benefit from some points of clarification (suggested below) and additional theoretical work on the MVK-oxide + DMS reaction to compare with the experimental results and contrast with the calculations on the $CH_2OO$ system. Please note that many of the comments in this review reflect the points that have already been raised in the thorough reviews of Rickard. Newland and Bloss, Blitz and the anonymous reviewer.

Main text

**Page 2, line 33**: The Welz et al. 2012 work is preceded by the Taatjes et al. JACS paper in which DMSO was used to generate the $CH_2OO$ Criegee intermediate.

AUTHORS' REPLY:

The reviewer is correct. However, the method reported by Taatjes et al.(Taatjes et al., 2008) is less efficient than that by Welz et al. (Welz et al., 2012) Nowadays, most photolytic generation of Criegee intermediates follow the method by Welz et al. The related sentences

*"However, due to their high reactivity and, hence, short lifetimes, laboratory studies of the reactions of CIs have been challenging. In fact, no direct detection of CIs has been known before Welz et al. reported a novel method to efficiently generate CIs other than through ozonolysis of alkenes (Welz et al., 2012)."*

would be modified to

*"However, due to their high reactivity and, hence, short lifetimes, laboratory studies of the reactions of CIs have been challenging until the work by Welz et al. who reported a novel*

*method to efficiently generate CIs other than through ozonolysis of alkenes (Welz et al., 2012)."*

**Page 2, line 41**: It is already established that ozonolysis experiments are by their very nature complicated – the authors should instead be more specific about the potential concerns they have regarding obtaining rate coefficients of Criegee intermediates from ozonolysis studies.

AUTHORS' REPLY:

We will clarify the situation by revising the related text to

*"Surprisingly, the obtained rate coefficients are up to $10^4$ times larger than previous results deduced from ozonolysis experiments (Johnson and Marston, 2008; Johnson et al., 2001; Hatakeyama and Akimoto, 1994). For ozonolysis experiments, typically only the ratios of reaction rate coefficients are obtained. The researchers have to compare with (at least) one absolute rate coefficient to get the rest rate coefficients. Unfortunately, the selected absolute rate coefficient (at that time) has large uncertainty, which propagates to other reported values. In addition, the reaction mechanism may be rather complicated and even the ratios of the rate coefficients need to be treated with care."*

**Page 2, line 51:** The very recent Cox et al. paper in ACPD (https://www.atmos-chemphys-discuss.net/acp-2020-472/) is also a thorough and up-to-date reference for existing studies of Criegee intermediate kinetics.

AUTHORS' REPLY:

Thanks. We will include this new reference. (Cox et al., 2020)

**Page 3, line 90:** A reference (or some further explanation) is needed regarding the MVKO precursor absorption at 308 nm.

AUTHORS' REPLY:

Below would be the measured absorbance of the diiodomethane (Exp# 12) and 1,3-diiodo-2-butene (Exp# 15) in the absorption cell (they are much diluted in the reactor cell). The absorption of 1,3-diiodo-2-butene at 308 nm is c.a. one-tenth of that at 248 nm. Consequently, we only perform the photolysis of 1,3-diiodo-2-butene at 248 nm.

[Figure]

**Page 4, line 110:** The authors should be able to determine an approximation of at least the MVK-oxide precursor concentration in their system. The vapor pressure of the precursor can be estimated using the Antoine coefficients. If the precursor was delivered to the reactor via a bubbler at a known flow rate, then the approximate concentration of the precursor can be deduced. In the event that the absorption coefficient of the precursor is deduced at a later date, this information would enable the concentration of MVK-oxide used in the present work to be obtained.

AUTHORS' REPLY:

Currently we don't have the available data for the cross section nor the empirical coefficients of Antoine coefficients for 1,3-diiodo-2-butene; hence we couldn't derive the absolute concentration. We have reported the deduced absorbance (*Abs*) of the precursor in the photolysis cell of different experiments sets in Table S3. The absolute concentration of precursor can be deduced from the *Abs* of precursor and other experimental conditions shown in Table S3., once the absolute cross section of 1,3-diiodo-2-butene is available.

We have modified the text to

*"However, because no absolute absorption cross sections for 1,3-diiodo-2-butene have been reported, its absolute concentration cannot be determined. We alternatively report the absorbance (Precursor Abs) of 1,3-diiodo-2-butene in the photolysis reactor (Table S3)."*

**Page 6, line 159:** Under the present experimental conditions, $CH_3$ would most likely undergo reaction with $O_2$ to form $CH_3OO$ and so it would be best to compare the reactivity of $CH_3OO$ (rather than $CH_3$) with I atom and Criegee intermediates.

AUTHORS' REPLY:

Thanks for pointing out. We will revise the sentences to

*"The expected products of DMS photolysis are $CH_3$ + $CH_3S$ (Bain et al., 2018). Under the presence of $O_2$ (10 Torr), $CH_3$ would be converted into $CH_3OO$. These radicals ($CH_3$, $CH_3OO$, and $CH_3S$) are less reactive than I atoms or CIs."*

**Page 7, line 205:** As the authors point out, there is currently significant uncertainty in the estimated and modelled steady state concentrations of Criegee intermediates. Because of this, it would be instructive to also frame the competitiveness of Criegee-initiated DMS oxidation vs. OH or NO3-initiated oxidation in terms of what concentration of Criegee intermediates are needed to oxidize a certain fraction (e.g. 5%, 10% or 20%) of atmospheric DMS using the theoretically determined rate coefficient.

AUTHORS' REPLY:

Possible concentrations of $NO_3$ and OH in the troposphere are found to be:

$[OH] = 1 \times 10^6$ $cm^{-3}$ (Li et al., 2018) and $[NO_3] = 10$ ppt $= 2.5 \times 10^8$ $cm^{-3}$ (Khan et al., 2015). Together with the reaction rate coefficients ($k_{DMS+OH} = 4.8 \times 10^{-12}$ $cm^3$ $s^{-1}$, $k_{DMS+NO3} = 6.8 \times 10^{-11}$ $cm^3$ $s^{-1}$ (Atkinson et al., 2004)), the concentration of CIs would have to be unreasonably high, at the order of $10^{11}$ $cm^{-3}$, to be competitive (5% of the effective reaction rate) with the DMS+OH and DMS+NO$_3$ reactions.

We would add the following sentences

*"If the DMS reactions with CIs were to be competitive to those with $NO_3$ (e.g., $2.5 \times 10^8$ $cm^{-3}$) and OH (e.g., $1 \times 10^6$ $cm^{-3}$) (e.g., 5% of the overall DMS removal), the concentration of CIs would have to be unreasonably high, at the order of $10^{11}$ $cm^{-3}$."*

**Page 8, line 220**. It seems peculiar that you have chosen to investigate theoretically only the CH2OO reaction and not the MVK-oxide reaction also. In MVK-oxide, the conjugation of the unsaturated side chain with the carbonyl oxide group has the potential to substantially

alter the surface. These calculations are likely significantly more complex than for the CH2OO case because of the need to consider syn and anti conformers, and cis/trans forms of each of these. However, given the interesting structural and conformeric dependence of Criegee intermediate reactivity, it is a regretful omission.

AUTHORS' REPLY:

Now we have the calculation result of MVKO+DMS reaction. Please see Reply to Referee 1 (for Line 220) for the calculation results on the direct reaction of MVKO + DMS, and the reply to referee 2 for catalysis reactions by DMS on unimolecular reactions of MVKO.

**Page 8, line 233**: A reference is needed to substantiate the statement regarding high barriers for DMS C-H insertion.

AUTHORS' REPLY:

We would add the paper of Decker et al. (**Phys. Chem. Chem. Phys.**, 2017,**19**, 8541-8551, doi:10.1039/C6CP08602K) in to the reference

**Page 8, line 251**: Do you anticipate stabilization of the (CH3)2SCH2OO adduct under tropospheric conditions?

The bonding is too weak to be stabilized under tropospheric conditions.

**Page 9, line 261:** You hypothesize that surface reactions converting DMS to SO2 in the chamber study of Newland could be the source of discrepancy between the present work and the work of Newland et al. I encourage the authors to respond to the comments of Rickard, Newland and Bloss, and Blitz regarding this matter.

AUTHORS' REPLY:

We would respond to the comments of Rickard et al. separately in the online discussion system of ACP.

**Figure 2**: Please include a note to address if the error bars are included or not included on this plot (as noted for Figure 4). Given that the rate coefficients for the self-reaction of CH2OO and the reaction of CH2OO + I (see Blitz review) are now well established, it would

be pertinent to deduce which of these is the major source of increased loss rates at higher laser fluence are under your experimental conditions.

AUTHORS' REPLY:

(i) We would add the following text in the caption:

*"For each data point, the error of the single exponential fitting is less than 1% (thus not shown)."*

(ii) Based on the absolute absorption cross section of $CH_2OO$ at 340 nm ($\sigma = 1.23 \times 10^{-17}$ cm$^2$) and the pressure-dependent yield of $CH_2OO$ from $CH_2I + O_2$ (0.46 at 300 Torr) (Ting et al., 2014a) the number densities of relevant species can be estimated to be the following (for Exp#1, Table S1).

$[CH_2OO]_0 = 6.7 \times 10^{11}$ cm$^{-3}$; $[I]_0 = 2.1 \times 10^{12}$ cm$^{-3}$; $[CH_2IOO]_0 = 7.7 \times 10^{11}$ cm$^{-3}$.

The first-order decay rate coefficient of $CH_2OO$ ($k_{eff}$) can be approximately estimated (Li et al., 2020) as:

$$k_{eff} = k_I[I]_0 + k_{self}[CH_2OO]_0$$

Using $k_{self} = 8 \times 10^{-11}$ cm$^3$ s$^{-1}$ and $k_I = 5.8 \times 10^{-11}$ cm$^3$ s$^{-1}$ at 300 Torr (Mir et al., 2020), the estimated $k_{eff}$ is 180 s$^{-1}$, consistent with the observed value of 232 s$^{-1}$ for $k_0$. Therefore, the main loss processes of $CH_2OO$ are reaction with iodine atoms (and other radicals) and its self-reaction.

In Figure S7, we can see a nice linear relationship between $k_0$ and the total produced radicals (proportional to the product of the laser fluence and the precursor concentration), further supporting the above mechanism. We would add the following sentences in the caption of Figure S7.

*"The main loss processes of $CH_2OO$ are reactions with radical byproducts like iodine atoms and its self-reaction. The observed values of $k_0$ (e.g., 232 s$^{-1}$ for Exp#1) are consistent with the values (e.g., 180 s$^{-1}$ at the condition of Exp#1) that are estimated using the reported kinetic data (yield and rate coefficients) (Mir et al., 2020; Ting et al., 2014)."*

**Supplementary information**

**Table S3:** Because both the reaction forming MVK-oxide from the precursor + O2 reaction

as well as the MVK-oxide + SO2 reaction features an adduct, the authors should label more caerefully the adduct referred to in the 'adduct yield' column of the table to avoid confusion.

AUTHORS' REPLY: We will add a footnote after the "adduct yield$^a$"

$^a$ *The yield of CH$_3$(C$_2$H$_3$)CIOO.*

**Figures S1, S2:** Provide details about error bars (c.f. comment about Figure 2).

AUTHORS' REPLY:

We would add the following text into the caption

*"For each data point, the error of the single exponential fitting is lees than 1% (thus not shown)."*

**Figure S4:** Please discuss the proposed origin of the "spike" at time zero.

AUTHORS' REPLY:

The photolysis of DMS produces radicals like CH$_3$ and CH$_3$S. A few vibronic bands of the A-X transition of the CH$_3$S radical (Liu et al., 2005) are within our probe window (335-345 nm). Thus it is possible that the "spike" near time zero is due to the absorption of the radical products of DMS photolysis, likely CH$_3$S or vibrationally excited CH$_3$S. We would add the following sentence in the figure caption.

*"The absorbance change under zero [DMS] comes from the interaction of the optics and the photolysis laser pulse, whereas the "spike" near time zero at high [DMS] may come from the absorption of the radical products of DMS photolysis, likely CH$_3$S (Liu et al., 2005) and/or vibrationally excited CH$_3$S."*

**S10:** These additional investigations are illuminating and interesting.

AUTHORS' REPLY: Thanks.

Additional comments regarding MVK-oxide conformers

I would like to add some discussion to the comments made by other reviewers regarding which conformers of MVK-oxide are produced from the photolytic scheme vs. ozonolysis. While the distribution of these conformers has not yet been deduced, the recent literature on

direct MVK-oxide kinetic and spectroscopic studies that indicated that both syn (Caravan et al., 2020) and anti (Vansco et al., 2020) confirmers are produced from the 1,3,-diiodobut-2-ene photolysis scheme used in the present work (Barber et al., 2018). Additionally, due to the rapid unimolecular decay of anti compared with syn (Barber et al., 2018;Vereecken et al., 2017), it is unlikely that reaction with DMS could compete with unimolecular decay under tropospheric conditions for the anti conformer.

AUTHORS' REPLY:

Same as the reply to Referee 2 (for Supplemental Information S20-S21), we have added some description to after line 80 to clarify the MVKO conformation.

"*For MVKO, there are 4 possible conformers. Following the nomenclature of Barber et al., syn/anti-MVKO (E/Z-MVKO) has a methyl/vinyl group at the same side of the terminal oxygen, while cis and trans refer to the orientation between the vinyl C=C and the carbonyl C=O bonds (Barber et al., 2018). It has been reported that syn- and anti-MVKO do not interconvert due to a high barrier between them but the barrier between cis and trans forms is low enough to permit fast interconversion at 298 K (Barber et al., 2018;Vereecken et al., 2017). Caravan et al., have shown that anti-MVKO is unobservable under thermal (298 K) conditions due to short lifetime and/or low yield, and thus, the UV-Vis absorption signal is from an equilibrium mixture of cis and trans forms of syn-MVKO (Caravan et al., 2020;Vereecken et al., 2017). For simplicity we will use MVKO to represent syn-MVKO (E-MVKO).*"

[revised manuscript text omitted]

[a] The yield of $CH_3(C_2H_3)CIOO$.

[b] The estimated absorbance of the precursor (1,3-diiodo-2-butene) at 238 nm in the photolysis reactor (using $L$ = 426 cm).

[c] Averaged value ± 1 sigma error of the mean (statistical only, not including systematic errors). The actual error bar would be larger since $k_r$ is highly correlated with other fitting parameters like (1−$\alpha$). Lin et al. have used $SO_2$ scavenger to obtain more robust results for $k_r$ (Lin et al., 2020).

[d] Standard deviation of the 15 data points of $k_{DMS}$.

**S2   Observed decay rate coefficient of CH₂OO at various conditions**

[revised manuscript text omitted]

lines, obtained with 248 nm photolysis laser ($I_{248nm}$ = 2.43 mJ cm$^{-2}$). See Exp#22 of Table S3 for the experimental conditions.

[Figure]

Fig. S6. Background traces under normal DMS concentrations, represented in colour lines, and the raw signal traces (without background subtraction), represented in grey lines, obtained with 308 nm photolysis laser ($I_{308nm}$ = 2.35 mJ cm$^{-2}$). See Exp#2 of Table S1 for the experimental condition. Note that the optics (longpass filters) are different from those at 248 nm.

**S6  Dependence of $k_0$, $k_{DMS}$, and $k_r$ on laser fluence and precursor concentration**

[Figure]

Fig. S7. Plot of $k_0$ against the product of the laser fluence ($I_{248nm}$ or $I_{308nm}$) and the precursor concentration [CH$_2$I$_2$] for the experiments (Exp#1−14, Tables S1−S2) of CH$_2$OO+DMS reaction. The x-axis essentially represents the total amounts of radical species generated through the photolysis of the precursor (R1) and the subsequent reactions (R2). Higher radical concentration results in faster CH$_2$OO decay, thus higher $k_0$. The difference of the slopes mainly comes from the difference of CH$_2$I$_2$ absorption cross sections at these two wavelengths (see Table S4). The main loss processes of CH$_2$OO are reactions with radical byproducts like iodine atoms and its self-reaction. The observed values of $k_0$ (e.g., 232 s$^{-1}$ for Exp#1) are consistent with the values (e.g., 180 s$^{-1}$ at the condition of Exp#1) that are estimated using the reported kinetic data (yield and rate coefficients)(Mir et al., 2020;Ting et al., 2014). Note that there are experiments having different combinations of [CH$_2$I$_2$] and $I_{308nm}$, but very similar $I_{308nm}$×[CH$_2$I$_2$] (like Exp#3,11; Exp#1,9).

[Figure]

Fig. S8. As Figure S7, but for the experiments (Exp#15−29) of MVKO+DMS reaction. Because the absorption cross section of the precursor (1,3-diiodo-2-butene) is not available, we use the absorbance at 238 nm in the reactor (using $L = 426$ cm) to represent the concentration.

[Figure]

Fig. S9. Plot of $k_{DMS}$ against the product of the laser fluence ($I_{248nm}$ or $I_{308nm}$) and the precursor concentration [$CH_2I_2$] for the experiments (Exp#1−14, Tables S1−S2) of $CH_2OO$+DMS reaction. The x-axis essentially represents the total amounts of radical species generated through the photolysis of the precursor (R1) and the subsequent reactions (R2). No observable trend of $k_{DMS}$ can be found for the data of 308 nm photolysis, whereas $k_{DMS}$ at 248 nm photolysis increases as $I_{248nm}$×[$CH_2I_2$] increases, which may result from the increased

radical generation from the DMS photolysis. Note that there are experiments having different combinations of [$CH_2I_2$] and $I_{308nm}$, but very similar $I_{308nm}$×[$CH_2I_2$] (like Exp#3,11; Exp#1,9).

[Figure]

Fig. S10. As Figure S9, but for $k_{DMS}$ in Exp#15−29 of MVKO+DMS reaction. No significant trend for $k_{DMS}$ is observed. Because the absorption cross section of the precursor (1,3-diiodo-2-butene) is not available, we use the absorbance at 238 nm in the reactor (using $L = 426$ cm) to represent the concentration.

[Figure]

Fig. S11. Plot of $k_r$ against the product of the laser fluence ($I_{248nm}$) and the absorbance of 1,3-diiodo-2-butene at 238 nm in the photolysis cell ($Abs$(238nm)) for the experiments of MVKO+DMS reaction (Exp#15−29, Tables S3). The x-axis essentially represents the total amounts of radical species generated through the photolysis of the precursor (R1) and the subsequent reactions (R2). Higher radical concentration results in faster decay of the adduct, thus higher $k_r$. Because the absorption cross section of the precursor (1,3-diiodo-2-butene) is not available, we use the absorbance at 238 nm in the reactor (using $L = 426$ cm) to represent the concentration.

**S7    Representative time traces for the CH₂OO+DMS reaction obtained with 308 nm photolysis**

[Figure]

Fig. S12. Representative time traces of CH₂OO absorption at 340±5nm at various [DMS] (Exp#1−4). The wavelength of the photolysis laser was 308 nm and the laser pulse is set at the time zero. In each experiment, [DMS] was scanned from 0 to the maximum (labeled as "up"), and scanned from maximum to 0 (labeled as "down"). The negative baseline is resulted from the depletion of the precursor CH₂I₂.

[Figure]

Fig. S13. As Fig. S12, but different experiment sets (Exp#5−8)

[Figure]

Fig. S14. As Fig. S12, but different experiment sets (Exp#9–11)

**S8 Representative time traces for the CH₂OO+DMS reaction obtained with 248 nm photolysis**

[Figure]

Fig. S15. Representative time traces of CH₂OO at 340±5nm at various [DMS] (Exp# 12−14). The wavelength of the photolysis laser was 248 nm and the laser pulse is set at the time zero. In each experiment, [DMS] was scanned from 0 to the maximum (labeled as "up"), and scanned from maximum to 0 (labeled as "down"). The negative baseline resulted from the depletion of the precursor CH₂I₂.

**S9    Representative time traces for the MVKO+DMS reaction obtained with 248 nm photolysis**

[Figure]

Fig. S16. Representative time traces of MVKO at 340±5nm at various [DMS] (Exp#15−18). The wavelength of the photolysis laser was 248 nm and the laser pulse is set at the time zero. In each experiment, [DMS] was scanned from 0 to the maximum (labeled as "up"), and scanned from maximum to 0 (labeled as "down").

[Figure]

Fig. S17. Representative time traces of MVKO at 340±5nm at various [DMS] (Exp#19–24). The wavelength of the photolysis laser was 248 nm and the laser pulse is set at the time zero.

[Figure]

Fig. S18. As Fig. S17, but different experiment sets (Exp#25–29)

**S10    Computational details for the reaction of CH$_2$OO + DMS**

**Additional methodological information**

The CH$_2$OO + DMS system was characterized at the CCSD(T)/aug-cc-pVTZ//M06-2X/ aug-cc-pV(T+d)Z level of theory. Though the computational demands of the reaction system prevent us from doing higher-level calculations at this time, this level of theory is expected to be sufficient to give a good idea of the PES layout, and derive rate coefficients with an accuracy of about one to two orders of magnitude. In particular, the study by Newland et al. proposes a very fast reaction which should then have a low energy barrier (Newland et al., 2015), whereas the current experimental study finds very slow elementary reactions which perforce must have a high energy barrier. The level of theory applied is able to easily discriminate between these extreme cases.

An additional set of exploratory calculations were performed at the M06-2X/cc-pVDZ level of theory, specifically on DMS + larger CI, DMS + CH$_2$OO in the presence of O$_2$, and unimolecular reactions of CH$_2$OO, *syn*-CH$_3$CHOO, *anti*-CH$_3$CHOO, and *cyc*-CH$_2$OOS(O)O- with and without complexation with DMS. These calculations at lower level are discussed here in the supporting information. For these calculations, only relative barrier heights on analogous reactions are important, which are sufficiently well described at the level of theory employed. As no indication for a significant enhancing effect on the reaction rate was found, no attempt was made to improve the absolute barrier height predictions.

**Impact of substitutions of the CI, or the presence of O$_2$, on the CI + DMS reaction**

Calculations for CH$_2$OO + DMS + O$_2$ reveal no influence of O$_2$ as a reaction partner, though the (CH$_3$)$_2$SCH$_2$OO adduct may form a complex with O$_2$ stabilized by few kcal mol$^{-1}$. O$_2$ addition on the (CH$_3$)$_2$SCH$_2$OO and CH$_3$S(=CH$_2$)CH$_2$OOH adducts, forming triplet peroxy radicals, was found to have large barriers exceeding 15 kcal mol$^{-1}$, and is not competitive against redissociation of the CI+DMS adducts even at atmospheric O$_2$ concentrations.

Calculations on the reactions of *syn*-CH$_3$CHOO and *anti*-CH$_3$CHOO with DMS show that, as opposed to the CH$_2$OO case, formation of (CH$_3$)$_2$SCH(CH$_3$)OO adducts is endothermic by a few kcal mol$^{-1}$, making reaction of substituted CI with DMS even less favorable. Finally, for all conformers of MVKO, the adduct with DMS was even found to be unstable at the M06-2X/cc-pVDZ level of theory: the needed C−S bond in the adduct appears to be too weak to compensate for the loss of conjugation stabilization in MVKO, and the system reverts to the MVKO + DMS complex instead, without a formal C−S bond. As a result, the barrier for the migration of a DMS methyl H-atom to the carbonyl oxide oxygen to form a methylidene adduct is ~10 kcal/mol higher than for the analogous TS in the CH$_2$OO+DMS system which does feature a weakly bonded intermediate adduct. The direct oxygen transfer from *E*- or *Z*-MVKO to DMS, forming MVK + DMSO, was found to have a similarly high energy barrier as in the CH$_2$OO+DMS system. No viable reaction channels were found

involving the double bond in MVKO. The lack of accessible transition states then prohibits rapid direct reaction between DMS and MVKO.

**DMS as a catalyst**

The experiments of Newland et al. found no evidence of DMS consumption (Newland et al., 2015), suggesting that the DMS activity hampering $SO_2$ oxidation by CI in their isoprene + $O_3$ system might be caused by catalytic effects. Hence, we examined whether the unimolecular decay of CI could be affected by complexation with DMS, performing a set of calculations using the lower level M06-2X/cc-pVDZ level of theory. At that level of theory, the complexes of $CH_2OO$, *syn*-$CH_3CHOO$ and *anti*-$CH_3CHOO$ with DMS are stabilized by 8.5 to 10.9 kcal mol$^{-1}$ (likely overestimated due to basis set superposition errors). The barriers for dioxirane formation in $CH_2OO$, *syn*-$CH_3CHOO$ and *anti*-$CH_3CHOO$ are 22.0, 25.8 and 18.4 kcal mol$^{-1}$ without DMS, respectively, while in the DMS complex they are 24.1, 26.8 and 20.4 kcal mol$^{-1}$ above the complex, respectively. In *syn*-$CH_3CHOO$, the energy barrier for 1,4-H-migration (vinylhydroperoxide channel) without and with DMS are 12.7 and 15.8 kcal mol$^{-1}$, respectively, again calculated from the bottom of the CI-DMS complex. The dominant unimolecular reaction of *E*-MVKO is a 1,4-H-shift (VHP-channel), where at the M06-2X/cc-pVDZ level of theory, we find similar results as for the methylated $CH_3CHOO$, i.e. the barrier height without (12.1 kcal/mol) and with complexing DMS (14.2 kcal/mol from the ground state of the complex) are essentially identical (see Table S5). For *Z*-MVKO, the dominant unimolecular reaction is a 5-membered ring closure, and here too, DMS does not affect the intrinsic energy barrier for the reaction (see Table S5).

These results, despite being at a less reliable level of theory, strongly suggest that the DMS complexation does not lower the intrinsic barriers for unimolecular rearrangements, and might even slightly increase them. Any catalytic effect of DMS on the unimolecular decomposition of CI is then due to the energy release of the complexation, but this is insufficient to lower the decay TS close to or below the energy level of free CI + DMS, such that the main fate of the complex remains redissociation without chemical loss. For example, the net energy barrier for the DMS-catalysed *Z*-MVKO unimolecular reaction is ~ +4 kcal/mol, still implying a slow bimolecular reaction. This is in agreement with the observations of the current experimental study, which sees no enhanced CI loss in the presence of DMS.

There are many other reactions in the isoprene + $O_3$ system that might be catalytically enhanced or slowed by DMS, and examining all of these is outside the scope of this study. We did examine the reaction of DMS with the adduct of $CH_2OO$ + $SO_2$, *i.e.* the thio-secondary ozonide (*cyc*-$CH_2OOS(O)O-$, thio-SOZ) (Kuwata et al., 2015;Vereecken et al., 2012) formed

prior to its decomposition to $SO_3$ + $CH_2O$. The DMS-catalyzed redissociation of thio-SOZ back to $CH_2OO$ + $SO_2$, thus inhibiting $SO_2$ oxidation by CI, was found at the M06-2X/cc-pVDZ level of theory to have an energy barrier of 17.8 kcal mol$^{-1}$, too high to compete against $SO_3$ formation for which a barrier $\leq 10$ kcal mol$^{-1}$ was found (Kuwata et al., 2015). Any inhibiting effect by DMS on the CI + $SO_2$ reaction is thus not caused by an enhanced redissociation of the thio-SOZ intermediate.

No data is available elucidating whether bimolecular reactions of CI-DMS complexes with suitable co-reactants ($SO_2$, $H_2O$, acids,…), or alternatively DMS complexes of such co-reactants with free CI, are hindered or enhanced relative to those of the free CI + co-reactant.

Table S5: ZPE-corrected DMS complex energies, E(complex), and barrier heights $E_b$ without and with a DMS complexing agent, at the M06-2X/cc-pVDZ level of theory. Energies are in kcal mol$^{-1}$ and relative to the free reactants.

| CI reaction | $E_b$ | E(complex) | $E_b$(complex) |
|---|---|---|---|
| $CH_2OO \rightarrow$ cyc-$CH_2OO$- | 22.0 | −9.6 | 14.5 |
| $Z$-$CH_3CHOO \rightarrow CH_2CHOOH$ | 12.7 | −8.6 | 7.2 |
| $Z$-$CH_3CHOO \rightarrow$ cyc-$CH(CH_3)OO$- | 25.8 | −8.6 | 18.2 |
| $E$-$CH_3CHOO \rightarrow$ cyc-$CH(CH_3)OO$- | 18.4 | −10.9 | 9.5 |
| Z-$(CH=CH_2)C(CH_3)OO \rightarrow$ cyc-$CH$-$CH_2C(CH_3)OO$- | 12.1 | −9.9 | 4.4 |
| $Z$-$(CH=CH_2)C(CH_3)OO$ + DMS $\rightarrow$ MVK + DMSO | 8.7 | | |
| $E$-$(CH_3)C(CH=CH_2)OO$ + DMS $\rightarrow$ MVK + DMSO | 8.0 | | |
| $(CH_3)C(CH=CH_2)OO$ + DMS $\rightarrow S(CH_3)(=CH_2)C(CH_3)(CH=CH_2)OOH$ | 11.2 | | |